# Dynamic organelle distribution initiates actin-based spindle migration in mouse oocytes

Xing Duan[1,2], Yizeng Li [3,4], Kexi Yi[5], Fengli Guo[5], HaiYang Wang[1,2], Pei-Hsun Wu [2], Jing Yang[4], Devin B. Mair[1,2,6], Edwin Angelo Morales[1], Petr Kalab [2], Denis Wirtz [2], Sean X. Sun [4] & Rong Li [1,2,7]*

Migration of meiosis-I (MI) spindle from the cell center to a sub-cortical location is a critical step for mouse oocytes to undergo asymmetric meiotic cell division. In this study, we investigate the mechanism by which formin-2 (FMN2) orchestrates the initial movement of MI spindle. By defining protein domains responsible for targeting FMN2, we show that spindle-periphery localized FMN2 is required for spindle migration. The spindle-peripheral FMN2 nucleates short actin bundles from vesicles derived likely from the endoplasmic reticulum (ER) and concentrated in a layer outside the spindle. This layer is in turn surrounded by mitochondria. A model based on polymerizing actin filaments pushing against mitochondria, thus generating a counter force on the spindle, demonstrated an inherent ability of this system to break symmetry and evolve directional spindle motion. The model is further supported through experiments involving spatially biasing actin nucleation via optogenetics and disruption of mitochondrial distribution and dynamics.

[1] Center for Cell Dynamics, Department of Cell Biology, Johns Hopkins University School of Medicine, 855 North Wolfe Street, Baltimore, MD 21205, USA. [2] Department of Chemical and Biomolecular Engineering, Whiting School of Engineering, Johns Hopkins University, Baltimore, MD 21218, USA. [3] Department of Mechanical Engineering, Kennesaw State University, Marietta, GA 30060, USA. [4] Department of Mechanical Engineering, Johns Hopkins University, Baltimore, MD 21218, USA. [5] Stowers Institute for Medical Research, 1000 East 50th Street, Kansas City, MO 64110, USA. [6] Department of Biomedical Engineering, Johns Hopkins University School of Medicine, Baltimore, MD 21205, USA. [7] Mechanobiology Institute, National University of Singapore, 5A Engineering Drive 1, Singapore 117411, Singapore. *email: rong@jhu.edu

The maturation of mammalian oocytes involves two consecutive asymmetric divisions that produce a large egg and two small polar bodies (PB). These asymmetric cell divisions depend on the eccentric positioning of the meiotic spindle adjacent to a polarized actomyosin-rich cortical domain[1–3]. Oocytes are maintained in the prophase of meiosis I (MI) and resume meiotic maturation in response to luteinizing hormone surge[4]. After germinal vesicle breakdown (GVBD), the first meiotic spindle is assembled in the center of the oocyte, which subsequently migrates to one side of the oocyte where an actomyosin-rich cortical domain is formed overlaying the MI spindle. Shortly after, PB extrusion occurs at this domain, which is accompanied by segregation of homologous chromosomes, resulting in the oocyte retaining the majority of the cytoplasm[5]. Different from mitotic cells[6], migration of the spindle toward the oocyte cortex is not dependent on microtubules, but rather, on actin filaments nucleated by the actin nucleators FMN2 and the Arp2/3 complex[7–10]. Our previous work characterized the process of spindle migration as a bi-phasic process: the initial movement of the MI spindle away from oocyte center requires FMN2[7,11]; once the spindle and its chromatin cargo are sufficiently close to the cortex, the chromatin-generated $Ran^{GTP}$ signal activates the Arp2/3 complex on the cortex, nucleating a cytoplasmic actin network, which drives cytoplasmic streaming to produce a hydrodynamic force that moves the spindle rapidly toward the cortex[12].

How FMN2-nucleated actin filaments (F-actin) initiates spindle migration from the oocyte center has remained a key unresolved question in the symmetry breaking process. FMN2 was originally identified as a gene product essential for female fertility in mice[9] and was later showed to be required for MI spindle migration[13]. Earlier work also showed that the function of FMN2 in spindle migration is dependent on its ability to nucleate F-actin[11,14]. Several models have been proposed regarding how FMN2 could drive spindle migration and therefore establish oocyte asymmetry[14–17]. Two most explicit force-generating models were proposed based on the observed FMN2 localization pattern: FMN2 localizes to both the oocyte cortex and the spindle periphery. One model proposed that cortical FMN2 nucleates actin cables that extend to MI spindle poles, and spindle migration results from one side winning a "tug of war" powered by myosin-II[14]. In a second model, we proposed that FMN2 nucleates F-actin at the spindle periphery to provide a local pushing force on the spindle[7]. Both models were proposed based on empirical observations and did not explain the mechanism for directional motion.

In this study, we first delineate the domains of FMN2 responsible for its targeting to either the oocyte cortex or the spindle peripheral region. Using domain-specific dominant-negative or FMN2 deletion constructs, we demonstrate that the spindle-peripheral, but not the cortical, localization of FMN2 is critical for MI spindle migration. We show that the spindle peripheral ER-associated FMN2 nucleates short actin bundles in a region surrounded by mitochondria. Using a mathematical model, we explore force production and symmetry breaking in the initiation of spindle motility. The model shows that the system can spontaneously break symmetry through dynamic actin assembly from the surface of ER-like vesicles and interaction of F-actin with nearby mitochondria. We then performed experiments to validate several predictions of the model.

## Results

### Role of different domains of FMN2 in MI spindle migration.
To determine whether the cortical or spindle-periphery localized FMN2 is critical for spindle migration, we first analyzed the

domains responsible for FMN2 localization. It was shown previously that the N-terminal 734 amino acids (aa) of FMN2 is responsible for its localization to the cortex and spindle periphery in mouse oocytes[11]. We divided the N-terminal aa 1–734 portion of FMN2 into several segments according to sequence complexity and known motifs and prepared DNA constructs of different N-terminal FMN2 fragments tagged with AcGFP (Fig. 1a). Expression of each fragment through mRNA injection in MI oocytes revealed that the aa 1–135 region of FMN2 is sufficient for FMN2 cortical localization, whereas aa 275–734 is sufficient for localization to the spindle periphery (Fig. 1b). Furthermore, $FMN2^{\Delta aa1-274}$ failed to localize to the cortex but only to the spindle periphery, whereas $FMN2^{\Delta aa136-734}$ is only found at the cell cortex (Fig. 1c). We therefore named FMN2 aa 1–135 cortical localization domain (CLD) and aa 275–734 spindle periphery localization domain (SLD) (see Fig. 1a).

To determine which pool of FMN2 is required for spindle migration, we first tested if CLD or SLD could serve as dominant negatives that interfere with the localization of full-length FMN2. Indeed, expression of CLD via mRNA injection disrupted the cortical localization of the co-expressed full-length FMN2 ($FMN2^{FL}$) tagged with AcGFP but left the spindle peripheral FMN2 pool largely unaffected (Fig. 1d). Conversely, overexpression of SLD abolished the spindle peripheral pool of FMN2 while having no effect on cortical FMN2 (Fig. 1d). To determine the effects of these dominant negative constructs on spindle migration, mouse oocytes with a centrally located germinal vesicle (GV) were microinjected with in vitro transcribed mRNA expressing either CLD or SLD. The oocytes were released from the cell cycle arrest to allow MI spindle formation and migration. After 14 h, the oocytes were fixed and scored for final spindle location. The percentage of oocytes in which spindle migration occurred was greatly reduced in SLD-expressing oocytes (23.19 ± 12.26%, n = 130), compared with the control (86.38 ± 3.63%, n = 106) or CLD mRNA-injected (74.57 ± 11.71%, n = 123) oocytes (Fig. 1e, f, Supplementary Movies 1–3). This was further confirmed with live imaging, which allowed for quantification of the distances of chromosomes movement over time and the migration speed (Fig. 1g, h).

We also tested the ability of FMN2 lacking each of the localization domains to rescue the spindle migration defects of $Fmn2^{-/-}$ oocytes. Both $FMN2^{FL}$ and $FMN2^{\Delta CLD}$ rescued the spindle migration defect of $Fmn2^{-/-}$ oocytes to similar levels, further supporting that cortical localization of FMN2 is not required for FMN2's function in spindle migration (Fig. 1i–l, Supplementary Fig. 1a, b and Supplementary Movies 4, 5). By contrast, in $FMN2^{\Delta SLD}$-injected $Fmn2^{-/-}$ oocytes, the spindle did not migrate (Fig. 1i–l, Supplementary Movies 6, 7). This result supports an essential role for SLD in spindle movement. Note that some of the SLD-injected WT oocytes, $Fmn2^{-/-}$ oocytes and $FMN2^{\Delta SLD}$-injected $Fmn2^{-/-}$ oocytes appeared to have imperfectly aligned chromosomes, however the spindle migration defect was not limited to these oocytes, and even in some wild-type oocytes the chromosomes also misaligned during spindle migration.

### Spindle peripheral FMN2 regulates local actin accumulation.
We previously showed that in MI oocyte F-actin concentrates both on the cortex and in the region surrounding the spindle, and the latter pool is fully dependent on FMN2[7]. FMN2 localizes to vesicularized ER concentrated around the spindle, shown previously by both fluorescence colocalization and immune-gold labeling[7]. Here, thin-sectioning electron microscopy analysis of oocytes at the MI spindle migration stage further showed that short bundles of actin filaments formed comet tail-like structures

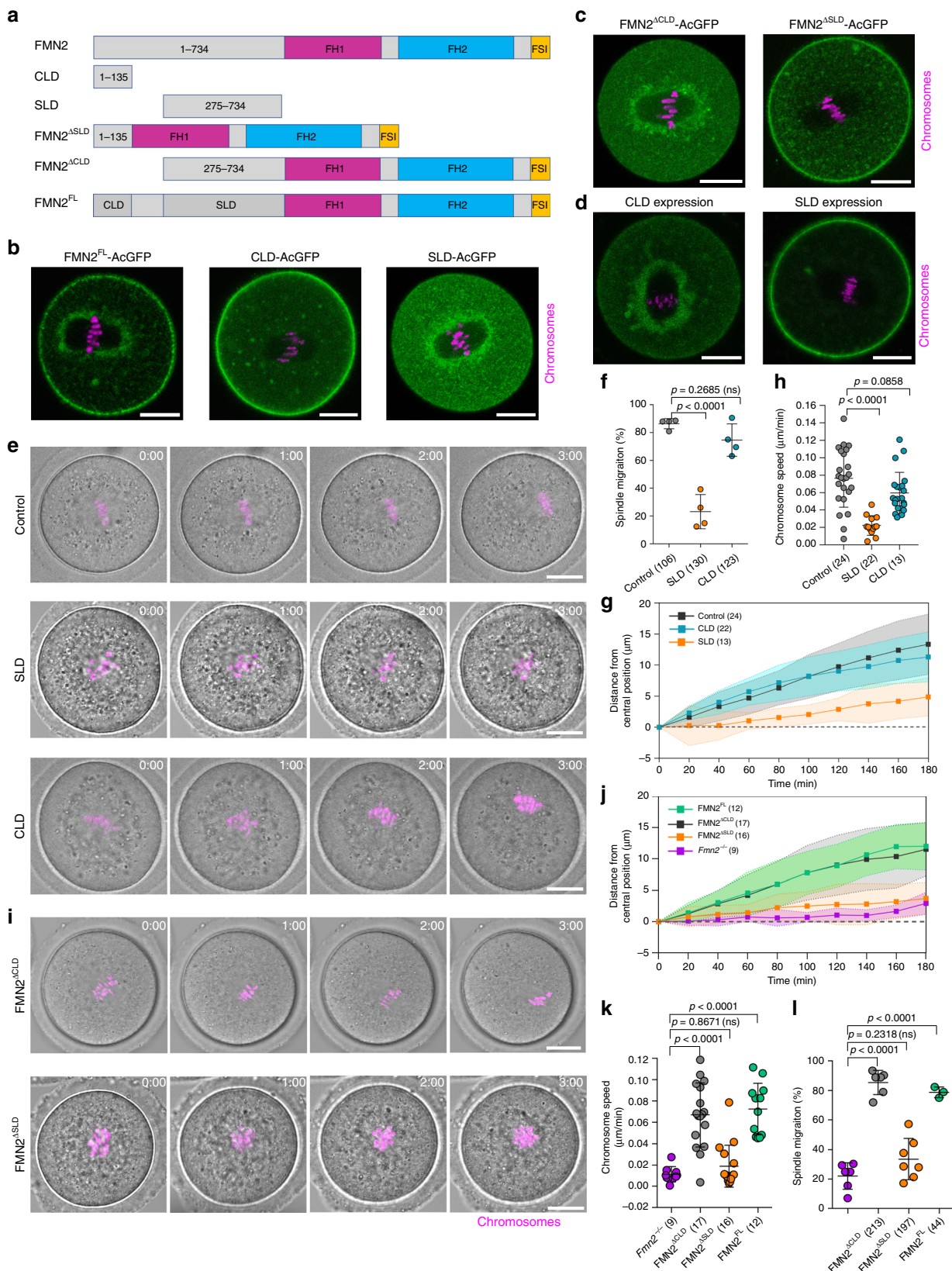

with end-on association with the surface of vesicles that are likely to be ER (Fig. 2a, Supplementary Fig. 3c, d), although the fixation condition in this experiment was incompatible with immunogold labeling. We therefore referred to these as ER-like vesicles. Note that these filaments are thinner (diameter: ~7 nm) and distinct from the intermediate filaments (~11 nm) and microtubules (~25 nm) that are present in oocyte cytoplasm (Fig. 2b). Cytoplasmic F-actin characterized by an enrichment around the spindle, as observed with fluorescent phalloidin, was disrupted in wild-type oocytes expressing SLD but not CLD (Fig. 2c, Supplementary

**Fig. 1 Spindle peripheral targeting of FMN2 is necessary and sufficient for MI spindle migration. a** Schematic diagrams showing various FMN2 constructs used in this study. FH1: formin homology domain 1; FH2: formin homology domain 2; FSI: formin–spire interaction. **b** Images showing the localization of full length FMN2 (green), aa 1–135 region of FMN2 (cortical localization domain: CLD) and aa 275–734 (spindle periphery localization domain: SLD) in wild-type oocytes. Scale bar, 20 μm. **c** Representative images (green) show the localization of FMN2$^{\Delta CLD}$ and FMN2$^{\Delta SLD}$ in wild-type oocytes. Representative results, $n = 23$ (left) and 7 (right). Scale bar, 20 μm. **d** Representative images show the effect of SLD and CLD expressions on full-length FMN2 localization. Representative results, $n = 5$ (left) and 20 (right). Scale bar, 20 μm. **e** Representative time-lapse images of spindle/chromosomes migration in wild-type oocytes or oocytes with SLD or CLD expression. Scale bar, 20 μm. **f** Quantification of the percentage of oocytes that underwent spindle migration after SLD and CLD expressions in wild-type oocytes from four independent experiments. **g** Chromosomes were tracked in oocytes as shown in **e** and the chromosomes movements were plotted. Data are represented as mean ± SD, from three independent experiments. **h** The speed of spindle/chromosomes migration was analyzed from the plots in **g**. Data are from three independent experiments. **i** Representative time-lapse images of chromosomes migration after FMN2$^{\Delta SLD}$ and FMN2$^{\Delta CLD}$ expression in $Fmn2^{-/-}$ oocytes. Representative results, $n = 16$ (upper) and 17 (lower). Scale bar, 20 μm. **j** The centroids of chromosomes were tracked and chromosomes movements were plotted as the distance from the initial central position over time. Data are from three independent experiments. **k** The speed of chromosomes migration was analyzed from the plots in **j**. **l** Quantification of the percentage of oocytes that underwent spindle migration after FMN2$^{\Delta SLD}$, FMN2$^{\Delta CLD}$, FMN2$^{FL}$ expression in $Fmn2^{-/-}$ oocytes. Data are from at least three independent experiments. All the data in this figure were analyzed by one side ANOVA, Tukey's multiple comparisons test. Data are represented as mean ± SD, and oocyte numbers are indicated in brackets. Source data are provided as a Source Data file.

Fig. 1c). Furthermore, FMN2$^{\Delta CLD}$ rescued the spindle-peripheral F-actin accumulation in $Fmn2^{-/-}$ oocytes (Fig. 2d, Supplementary Fig. 1d), whereas expression of FMN2$^{\Delta SLD}$ did not but caused an increase in cortical F-actin level (Fig. 2d, Supplementary Fig. 1d). These results demonstrate that the effects of the expression of various FMN2 mutants on spindle migration correlate with their ability to support actin polymerization at the spindle periphery.

**Distribution of mitochondria during MI spindle migration.** Electron microscopy images also showed that, roughly outside of the layer of ER-like vesicles, a region of mitochondria enrichment (shown in purple in Fig. 3a) encircles the ER-like vesicles and spindle. These two organelle regions are not fully separated but infiltrate each other (Fig. 3a). This was of interest because the proximity of these organelles could allow mitochondria to serve as dynamic barriers against which actin filaments polymerizing from the surface of ER-like vesicles produce a pushing force on the spindle. We therefore used fluorescence microscopy to further characterize the distribution of mitochondria during MI. Before GVBD, mitochondria were dispersed in the cytoplasm. After GVBD and while the spindle was being assembled during a 2–3 h time window, mitochondria gradually accumulated around the spindle (Fig. 3b and Supplementary Fig. 2b, Supplementary Movie 8). This spindle peripheral accumulation of mitochondria was largely disrupted in $Fmn2^{-/-}$ oocytes or oocytes injected with SLD but not CLD mRNA (Fig. 3c–f, Supplementary Fig. 2f). There was no difference of mitochondrial cluster size between SLD-expressing oocytes and $Fmn2^{-/-}$ oocytes (Supplementary Fig. 2g). Expression of the mutant protein FMN2$^{IRK}$ (mutation of residues Ile 1215, Arg 1295, and Lys 1371 in the formin homology domain 2 (FH2 domain) to Ala, which specifically disrupt the actin nucleation activity of FMN2[11]) in $Fmn2^{-/-}$ oocytes, did not rescue the defect in mitochondria accumulation around the spindle (Fig. 3g and Supplementary Fig. 2c). Consistently, depolymerization of cytoplasm actin by Latrunculin A (LatA) before GVBD also disrupted the spindle-peripheral accumulation of mitochondria (Supplementary Fig. 2d, e). As expected, expression of FMN2$^{\Delta CLD}$, but not FMN2$^{\Delta SLD}$, rescued the spindle-peripheral mitochondria distribution in $Fmn2^{-/-}$ oocytes to a similar degree as FMN2$^{FL}$ expression (Fig. 3e, g). These data suggest that the spindle-peripheral FMN2 and its actin nucleation activity are important for the accumulation of mitochondria in this region.

Myosin Vb, an actin-based motor implicated in the motility of many organelles, such as ER and mitochondria[18–20], is known to move along actin filaments nucleated by formin proteins in

mouse oocytes and other cell types[15,21–24]. To test if myosin Vb is involved in mitochondria accumulation at the spindle periphery, a dominant negative construct comprising the tail domain of myosin Vb[15,25], was expressed in oocytes via mRNA injection. Oocytes expressing myosin Vb tail domain showed a dispersed mitochondrial distribution with reduced accumulation in the spindle periphery (Fig. 3h, i). By contrast, myosin Vb tail expression did not cause mis-localization of ER (Fig. 3j) or FMN2 (Fig. 3k). Previous work showed that expression of myosin Vb tail affected the distribution of Rab11a labeled endosomes[26,27]. We also observed co-localization of myosin Vb tail with GFP-Rab11a foci, however, the speed of movement of these foci was not significantly altered compared to that in control oocytes (Supplementary Fig. 2h, j). Moreover, we directly tethered Myosin Vb tail to mitochondria through a mitochondrial outer membrane-binding motif (ActA)[28]. This construct strongly dispersed mitochondria from the spindle periphery to the cortex (Supplementary Fig. 2k). The GFP-Rab11a-labeled endosomes were not co-localized with Myosin Vb tail-ActA in the cytoplasm and the speed of GFP-Rab11a-labeled endosomes was also unaffected (Supplementary Fig. 2i, j). Thus, actin and myosin Vb-mediated transport may play a direct role in mitochondria accumulation around the spindle in oocytes, but detailed mechanisms remain to be elucidated.

Time-lapse confocal imaging and quantification showed that, during MI spindle migration, ER, as labeled with mKate2-tagged Sec61β (also confirmed with GFP-VapA, BFP-KDEL, or DiI[29,30], Supplementary Fig. 3a, b), remained closely associated with the spindle, with a distribution that became slightly asymmetric during spindle migration (Fig. 3l, m, Supplementary Fig. 3e and Supplementary Movie 9). On the other hand, upon the start of MI spindle migration, the distribution of mitochondria changed from symmetric to strongly asymmetric with more mitochondria surrounding the back half of the spindle, while the front of the spindle became devoid of mitochondria (Fig. 3l, m; Supplementary Fig. 4a–f and Supplementary Movie 9). The changes in the distribution of these organelles was also observed in a previous study[31]. This observation suggests that mitochondria, but not ER, experience forces opposite to the force that drives spindle movement.

**A biomechanical model of MI spindle migration.** We implemented a biomechanical model for spindle migration in MI based on actin polymerization in the spindle periphery and dynamic organization of the ER and mitochondria (see section "Methods" for model details). We assume that actin filaments are nucleated from ER vesicles tethered to the spindle surface. The

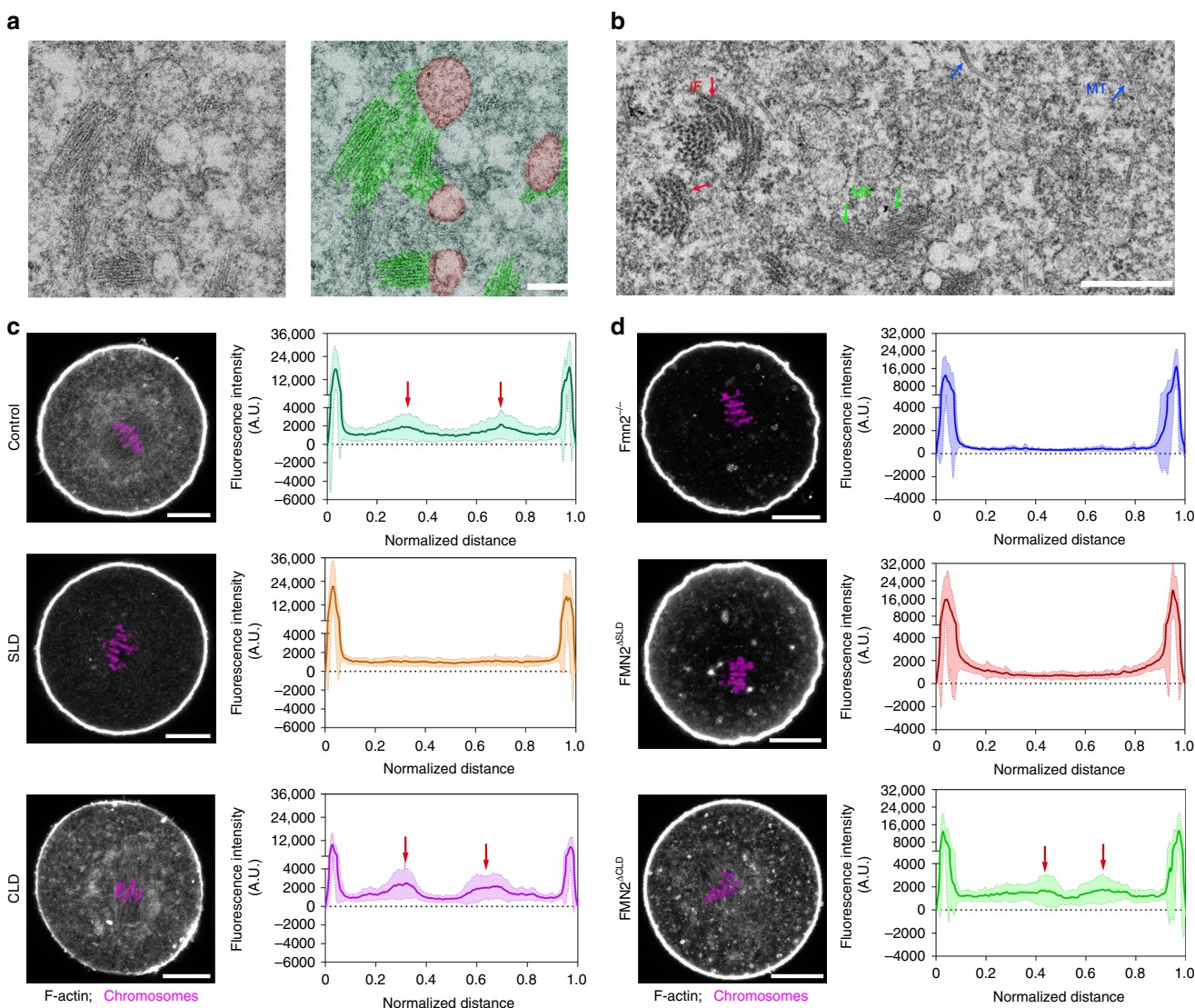

**Fig. 2 Spindle peripheral FMN2 is required for actin polymerization surrounding the spindle. a** Representative EM images of MI oocytes show that short bundles of actin filaments form comet tail-like end-on association with the surface of ER-like vesicles. Red, ER-like vesicles. Green, F-actin bundles. Representative results, $n = 3$. Scale bar, 0.2 μm. **b** Representative EM images of MI oocytes show different appearance and thickness of microfilament (MF, green arrow), intermediate filament (IF, red arrow) or microtubule (MT, blue arrow). Representative results, $n = 3$. Scale bar, 0.5 μm. **c** Representative images of rhodamine phalloidin staining of F-actin after SLD or CLD expression in wild-type oocytes. The accumulation of F-actin surrounding the spindle was disrupted in oocytes expressing SLD but not CLD. The curves show the average of fluorescence intensity trace of actin along a thick line crossing the mid-zone of the MI spindle and the shades show the SD of the central lines (see Supplementary Fig. 1c). Red arrows point to actin intensity peaks around the spindle. $n = 21$ (control), $n = 13$ (CLD), $n = 21$ (SLD). Scale bar, 20 μm. **d** Representative images of phalloidin staining in $Fmn2^{-/-}$ oocytes with FMN2$^{\Delta SLD}$ or FMN2$^{\Delta CLD}$ expression. FMN2$^{\Delta CLD}$ rescued the spindle peripheral F-actin accumulation in $Fmn2^{-/-}$ oocytes. Line traces of average fluorescence intensity of actin and SD (shade) are shown as in **c**. $n = 19$ ($Fmn2^{-/-}$), $n = 15$ (FMN2$^{\Delta CLD}$), $n = 17$ (FMN2$^{\Delta SLD}$). Scale bar, 20 μm.

ER-associated sites of actin nucleation are initially randomly distributed along the periphery of the spindle. Mitochondria (Mitos), represented as interacting spherical bodies (referred to as Mitos in the model), are distributed immediately outside of the region of actin nucleation (Fig. 4a). As we do not fully understand the processes governing mitochondria morphology and distribution in oocytes, we consider mitochondria around the spindle as a fluid-like structure with some cohesion, while also allowing for separation and diffusion. The simplest model that captures these features is a system of particles with short-range attraction and repulsion. Attraction tends to pull particles together, which may represent mitochondrial fusion—a process that involves attractive molecular forces exerted through mitofusin oligomerization, whereas repulsion tends to separate particles,

which may approximate mitochondrial fission mediated by constricting dynamin helices[32]. Polymerization of F-actin extends actin bundles toward mitochondria, and upon contact results in reaction forces from the mitochondria that push the spindle, but these forces are initially distributed randomly around the spindle. We then performed simulations using, as much as possible, parameters estimated based on experimental observations to test whether the system could break symmetry and produce a net directional force that moves the spindle (Fig. 4a–c) (also see the "Methods" section and Supplementary Table 1 for details and parameters used).

Figure 4d shows the trajectory of the center of the spindle for a typical simulation. Initially, the distributed force leads to random motion of the spindle, and mitochondria remain uniformly

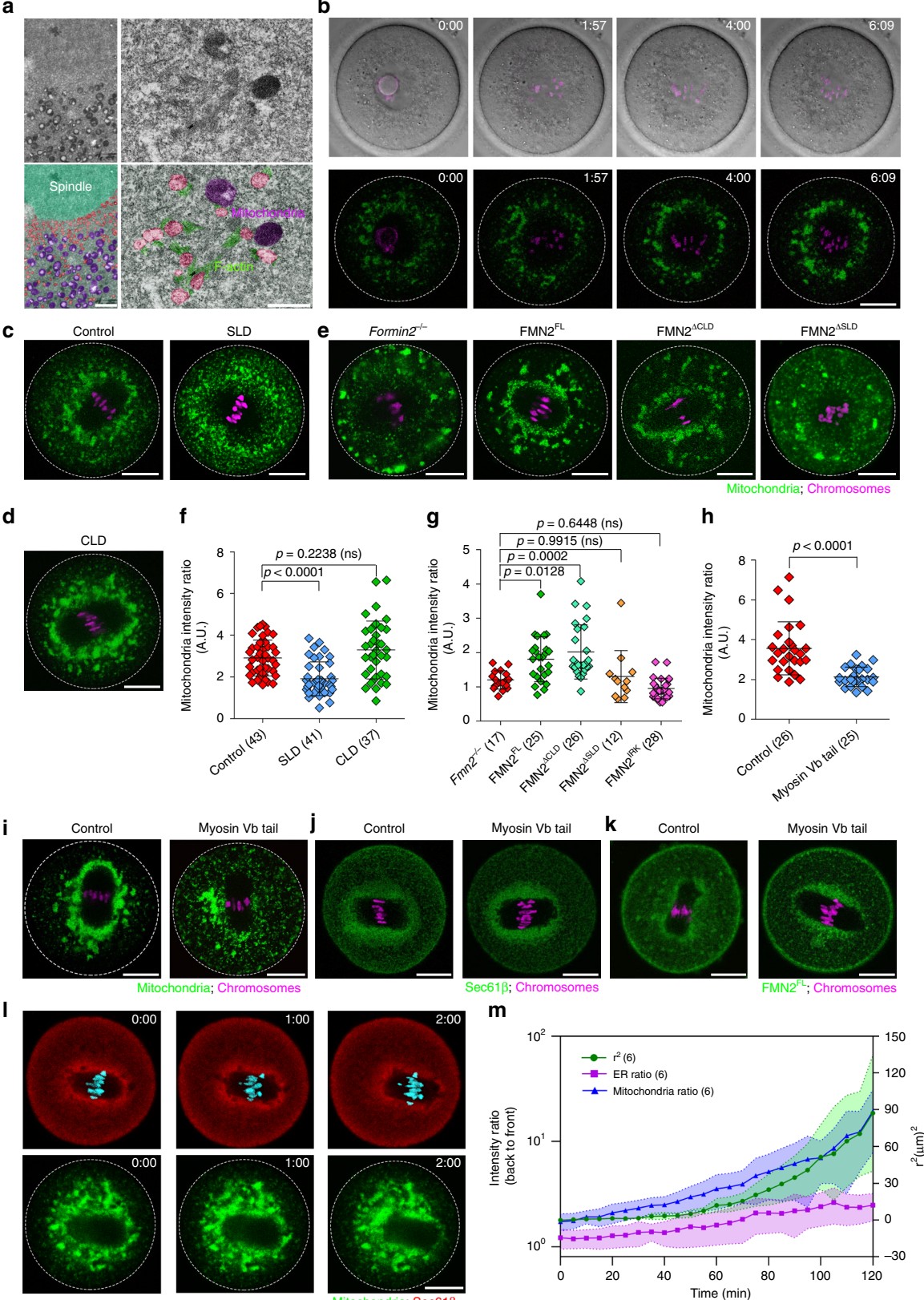

distributed around the spindle (Fig. 4e). The distribution of mitochondria gradually becomes asymmetric (Fig. 4f), and soon after, spindle translocation initiates with the leading end of the spindle surrounded by fewer mitochondria than the lagging end (Fig. 4g, Supplementary Fig. 5b, and Supplementary Movie 10).

This is intuitive: a higher concentration of mitochondria at the back end of the spindle allows for higher production of pushing forces on the spindle than the end with less mitochondria, and these mitochondria and thus force asymmetry is in turn further exaggerated by spindle movement. The co-evolution of

**Fig. 3 Dynamic distribution of mitochondria during MI spindle migration. a** Electron microscopy images showing the concentration of mitochondria immediately outside the ER-like vesicles zone surround the spindle. Red, ER-like vesicles. Purple, mitochondria, Green, actin filaments. $n = 3$. Scale bar, left: 2 μm; right: 0.5 μm. **b** Representative time-lapse DIC images (upper panels) and images of mitochondrial distribution (low panels) during and after GVBD in wild-type oocyte. $n = 8$. Scale bar, 20 μm. **c**, **d** Representative images of mitochondria distribution in control ($n = 43$), SLD-expressing (**c**, $n = 41$) and CLD-expressing (**d**, $n = 37$) oocytes. Scale bar, 20 μm. **e** Representative images of mitochondria distribution in $Fmn2^{-/-}$ oocyte ($n = 17$) without or with the expression of $FMN2^{FL}$ ($n = 25$), $FMN2^{\Delta CLD}$ ($n = 26$), and $FMN2^{\Delta SLD}$ ($n = 12$). Scale bar, 20 μm. **f** Quantification of mitochondria accumulation around spindle (mitochondria at spindle periphery/mitochondria outside spindle periphery) in control, SLD and CLD-expressing wild-type oocytes. Data are from four independent experiments. **g** Quantification of mitochondria accumulation around the spindle in $Fmn2^{-/-}$ oocyte, or $FMN2^{FL}$, $FMN2^{\Delta CLD}$, $FMN2^{\Delta SLD}$, $FMN2^{IRK}$-expressing $Fmn2^{-/-}$ oocytes. Data are from three independent experiments. **h** Quantification of mitochondria accumulation around the spindle in control and myosin Vb tail-expressing oocytes. Data are from four independent experiments. Two-tailed unpaired $t$-test. **i** Representative images of mitochondria distribution in control ($n = 26$) and myosin Vb tail-expressing ($n = 25$) oocytes. Scale bar, 20 μm. **j** and **k** Representative images of ER (**j**, $n = 17$) and FMN2 (**k**, $n = 12$)) localization in control and myosin Vb tail-expressing oocytes. Scale bar, 20 μm. **l** Representative time-lapse images of ER and mitochondria distribution during spindle migration in wild-type oocytes. Scale bar, 20 μm. **m** Quantification of mitochondria and ER distribution during spindle migration. Green line: $r^2$; Blue line: mitochondria ratio; Magenta line: ER ratio. SD was shown by the shade. The data in Fig. 5f, g was analyzed by one-side ANOVA, Tukey's multiple comparisons test. Data are represented as mean ± SD. Source data are provided as a Source Data file. Oocyte numbers are indicated in brackets.

mitochondria asymmetry and directional spindle movement is also observed experimentally (Fig. 3m, Supplementary Movie 9).

**Experimental validation of the biomechanical model.** We tested the biomechanical model first by exploring the parameter space describing mitochondrial dynamics required for the breaking of mitochondrial symmetry and directed spindle migration (Supplementary Fig. 5a–i). Our analysis showed that an appropriate ratio of repulsion to attraction among mitochondria is important for spindle movement. When the repulsive force among mitochondria becomes too high compared to the attractive force, mitochondria are dispersed from the spindle periphery and directional spindle migration could not be initiated (Fig. 4h, Supplementary Movie 11). Likewise, if the attractive force is too high relative to the repulsive force, mitochondria tightly cluster around the spindle and symmetry breaking also could not occur (Fig. 4i, Supplementary Movie 12). To test these predictions, we reduced mitochondrial fission by knocking down $Drp1$ or increased mitochondrial fusion by overexpression of Mitofusin 1 ($MFN1$)[33], to mimic the scenario of high attractive force relative to repulsive force. Indeed, partial $Drp1$ knockdown or overexpression of MFN1 resulted in tight mitochondria clustering around the spindle. Intracellular ATP and calcium levels were not obviously affected, as shown by using an ATP biosensor and Fluo-4 calcium indicator except that overexpression of MFN1 slightly increased the ATP content (Fig. 4j, Supplementary Figs. 6a–d, 7a, b). In these oocytes spindle migration failed, and mitochondria retained a tight and even distribution around the spindle (Fig. 4j–m, Supplementary Movies 13, 14, 16). To test the effect of reducing the accumulation of mitochondria around the spindle, we examined spindle migration in oocytes injected with mRNA encoding myosin Vb tail or Myosin Vb tail-Acta. A significant reduction in spindle migration was observed in the myosin Vb tail or myosin Vb tail-Acta-injected oocytes without any effect on intracellular ATP and calcium levels, compared with the control (Fig. 4j–m, Supplementary Fig. 6c, d and Supplementary Movies 15, 17), further supporting the importance of spindle-periphery accumulation of mitochondria in spindle movement, although we could not completely rule out the involvement of other organelles whose distribution is also regulated by myosin Vb.

Finally, the model predicts that if the initial distribution of actin nucleation is asymmetric between the two spindle halves, the side with more nucleation would predictably be the trailing end (Fig. 5a, Supplementary Movie 18). To test this, we developed an optogenetic system to recruit the C-terminal half of FMN2 (FMN2-C), responsible for actin nucleation, to the ER associated

with one side of a centrally located and not-yet-moving spindle, through the ER membrane protein Sec61β. Specifically, the miLID domain[28] was fused to the N-terminus of EYFP-Sec61β, which symmetrically localizes to spindle periphery; local photo-activation of the miLID by 488 nm light leads to recruitment of FMN2-C fused to mCherry-mSSPB[R73Q] to one pole of the spindle (Fig. 5b, c). Time-lapse imaging showed that recruiting FMN2-C to one side of the spindle always induced spindle movement with the opposite pole leading (21/23 oocytes observed; the other two did not show spindle migration) (Fig. 5c, e, Supplementary Movie 19). For control, the same light treatment was applied to oocytes expressing mCherry-mSSPB[R73Q] without the fusion with FMN2-C resulted in roughly equal frequency for each end to lead the spindle movement (Fig. 5d, e, Supplementary Movies 20, 21). In another experiment with light-induced recruitment of FMN2-C-mCherry-mSSPB[R73Q] to EYFP-Sec61B-miLID at one of the spindle poles, after the spindle-initiated apparent movement, we switched light activation to the opposite pole—the one that was initially leading the movement (Fig. 5f). In 5 out of 7 cases, the spindle first stopped moving and then started moving in the opposite direction (Fig. 5f–h, Supplementary Movie 22). In the other two oocytes, the second light activation was unable to stop the movement caused by the first activation. These results strongly support the model whereby FMN2 nucleated actin from the surface of ER-like vesicles generates the pushing force to drive spindle movement.

**Discussion**

The results described above provide mechanistic insights into how FMN2-nucleated actin filaments drive the first phase of spindle movement and the initiation of symmetry breaking. Our demonstration that the spindle peripheral FMN2 and its associated actin assembly, but not the cortically localized FMN2, are necessary and sufficient for spindle migration is consistent with the notion that the initial movement of the spindle is driven by an actin-based pushing force in the proximity of the spindle[7]. Our optogenetic experiments controlling the asymmetry of FMN2-mediated actin nucleation and biasing the direction of spindle migration provided a more direct support to the notion that the spindle is being pushed. This result does not support the model in which spindle movement is due to pulling forces of contractile actin bundles generated by FMN2 connecting the cortex with spindle poles[14] or the model, whereby the spindle is moved by an actin network formed from evenly distributed actin nucleation sites in the cytoplasm[27]. Both these alternative models would predict spindle movement toward the side in which actin

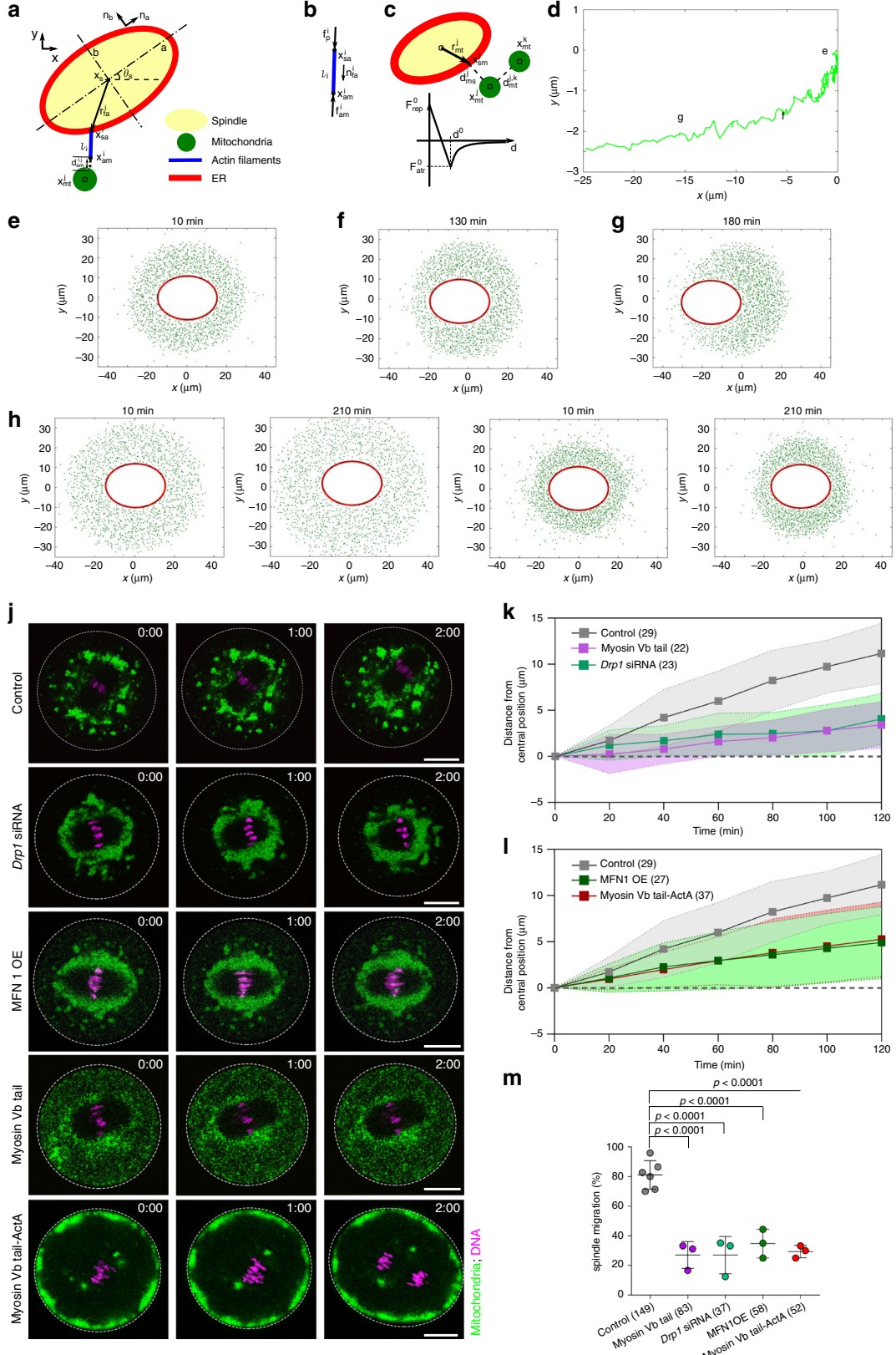

assembly is enhanced. In addition, during spindle migration, mitochondria move in the direction opposite to spindle movement and accumulate to the back half of the spindle, suggesting that mitochondria are experiencing a counter force. When the spindle periphery mitochondria were tightly clustered by *Drp1* knockdown or MFN1 overexpression, spindle migration was prevented. These results collectively support our model whereby FMN2-nucleated actin filaments push against mitochondria to produce a pushing force on the spindle.

Aside from the mechanism of force production, none of the previous models offered an explicit explanation of how symmetry may be broken to initiate directed motion of the MI spindle

**Fig. 4 A Mathematical model of MI spindle migration. a** Illustration of the coordinate system of the spindle, actin filaments, and mitochondria. The ER, which carries FMN2, is assumed to attach to the spindle at all time. More details of the model can be found in the section "Methods". **b** Force balance on each actin filament. **c** Top: illustration of the repulsion and attraction force among mitochondria and between the spindle and mitochondria. Bottom: Diagram of the repulsion and attraction force as a function of distance *d*. **d** Trajectory of the center of the spindle in the control case. **e–g** Simulations showing the relative position of the spindle and the mitochondria distribution at three different time points as marked in the trajectory in **d**. **e** Initial state, **f** starting migration, **g** mid-way of migration. Note that due to their very short length, actin filaments are not visible in these representations. **h** Model for myosin Vb tail expression. **i** Model for *Drp1* knockdown. **j** Representative time-lapse images of mitochondria distribution during spindle migration in control siRNA-injected (*n* = 29), *Drp1* siNRA-injected (*n* = 23), myosin Vb tail-expressing (*n* = 22), MFN1 over-expressing (OE) (*n* = 27), or Myosin Vb tail-Acta-expressing oocytes (*n* = 37). Green, mitochondria. Magenta, chromosomes. Scale bar, 20 μm. **k, l** Centroids of chromosomes were tracked in oocytes and the movement of chromosomes position were plotted in *Drp1* siRNA-injected, myosin Vb tail-expressing (**k**), MFN1 OE, or myosin Vb tail-Acta-expressing (**l**) oocytes. Data are represented as mean ± SD, from three independent experiments. Oocyte numbers are indicated in brackets. **m** Quantification of the percentage of oocytes that underwent spindle migration for WT oocyte, *Drp1* siRNA-injected, MFN1OE, myosin Vb tail-expressing, and myosin Vb tail-ActA-expressing oocytes. See Supplementary Fig. 6a, b for the efficiency of *Drp1* knockdown. Data are represented as mean ± SD. Data are from three independent experiments. One side ANOVA, Tukey's multiple comparisons test. Data are represented as mean ± SD. Oocyte numbers are indicated in brackets. Source data are provided as a Source Data file.

toward the cortex. Our model suggests that symmetry breaking results effectively from a positive feedback loop between the force exerted on the spindle, which is dependent on the presence of mitochondria crowding in the region of actin polymerization, and the effect of spindle movement on mitochondrial distribution. When mitochondrial distribution around the spindle becomes asymmetric with an accumulation to the back half of the spindle, the pushing forces on the spindle become vectoral, allowing the spindle to translocate in the direction of the leading pole. On the other hand, movement of the spindle in the leading pole direction further enhances the mitochondrial distribution asymmetry. The model predicts co-emergence of mitochondria distribution asymmetry and directed spindle motion, which is consistent with experimental observations.

The finding of this study also modifies our previous conclusion of bi-phasic spindle movement[7]. We proposed that the FMN2-dependent first phase of movement was purely a random walk due to random forces on the spindle, and symmetry breaking results mostly from Arp2/3 activation at the cortical pole proximal to the chromatin and the ensuing cytoplasmic streaming. Our present model suggests that the FMN2-based system of actin assembly at the spindle periphery is also capable of symmetry breaking and could play a role in the initiation of migration in a persistent direction. Interestingly, our model also predicts that the FMN2-based pushing mechanism has an inherent limit in the maximal distance through which the spindle could move, which is roughly half of the spindle length (see the "Methods" section). Because the length of MI spindle is ~20 μm and the radius of an MI oocyte is ~35 μm, the FMN2-based system might be insufficient to deliver the spindle all the way to the cortex, thus necessitating the Arp2/3 complex-mediated force near the cortex.

An unexpected component of the FMN2-based pushing mechanism is mitochondria. It is known that a wave of mitochondrial biogenesis occurs at the start of oocyte maturation[34–38]. While the concentration of mitochondria surrounding the spindle could be important for locally supplying the ATP that powers spindle formation and movement, our finding suggests a mechanical role for mitochondria in providing the counter balance for the polymerizing actin filaments to exert force on the spindle. Furthermore, symmetry breaking and the generation of a net unidirectional force to drive spindle movement requires dynamic organization of mitochondria through balanced fusion and fission, as well as sufficient concentration of mitochondria around the spindle. Since mitochondrial abnormalities have been associated with age-related reproductive decline[39,40], it will be interesting to investigate in future work whether defects in mitochondria dynamics and organization related to spindle migration may be a cause of the reduced egg number and quality during female reproductive aging.

## Methods

**Mouse oocyte preparation and culture**. All mouse care and use were approved by the Institutional Animal Care and Use Committee at the Johns Hopkins University. Mice were housed with free access to food and water under a 12-h light/12-h dark cycle. Mice were fed a diet containing low fiber (5%), protein (20%), and fat (5–10%). Mouse rooms were maintained at 30–70% relative humidity and a temperature of 18–26 °C (64–79 °F) with at least 10 room air changes per hour. Euthanasia of mice was performed by carbon dioxide asphyxiation, followed by cervical dislocation. GV oocytes were collected from 8 to 10 weeks old female CF1 mice or 12–15 weeks old female 129/Sv *Fmn2*$^{-/-}$ mice[9] after injecting with pregnant mare serum gonadotropin (5 IU; Sigma) and cultured in M16 medium containing 0.2 mM 3-isobutyl-1-methylxanthine (IBMX) (Sigma-Aldrich) after microinjecting with mRNA encoding fluorescently labeled proteins under mineral oil at 37 °C in a 5% $CO_2$ atmosphere. To examine the mitochondria distribution after Lat A treatment, oocytes were treated with Lat A (Millipore, 428026) right after collection and then incubated with IBMX before meiotic resumption overnight (10 h) and also during spindle formation. In some experiments, oocytes were treated with 5 μg/ml oligomycin (Sigma, O4876), 5 μM BAPTA-AM (provided by Takanari Inoue) or stained with 5 μM Fluo-4 (Thermo Fisher, F14201).

**Plasmid construction and mRNA transcription**. The N-terminal domain of FMN2 was divided into several segments according to the sequence complexity and known motifs. Different mouse formin 2 (Fmn2) constructs as indicated in Fig. 1a were inserted into the pGEMHE vector (subcloned from pGEMHE-H2B-mCherry plasmid obtained from Jan Ellenberg lab[41]) in frame with AcGFP. FMN2-mCherry[14], pGEMHE-mCherry-full length myosin Vb[15], and pGEMHE-mCherry-myosin Vb tail constructs[42] were provided by Jan Ellenberg (European Molecular Biology Laboratory, Heidelberg, Germany), Marie-Hélène Verlhac (Collège de France, CNRS-UMR7241, INSERM-U1050, Paris, France) and Melina Schuh (Department of Meiosis, Max Planck Institute for Biophysical Chemistry, Germany). Sec61β ORF (provided by T. Rapoport, Harvard Medical School, Boston, MA, USA) was inserted into pGEMHE-mKate2. Mitochondrial targeting sequence[43] (Addgene, 23348) was inserted into pGEMHE vector in frame with AcGFP. For a red mitochondria label, Por1-mCherry was inserted into pYX-RNA-EGFP vector (a derivative of pYX-Asc1[44], provided by Petr Solc, IAPG, Czech Republic). Mitochondrial outer membrane-binding motif (Acta) was from Addgene (60413)[28]. The BFP-KDEL plasmid[30] was from Addgene (49150) and subcloned to pGEMHE vector. Mitofusin1[45] (Addgene, 23212), GFP-Rab11a (Addgene, 12674)[46] and ATeam1.03-nD/nA ORF[47] (Addgene, 51958) were inserted into pGEMHE vector. The miLID and mSSPB$^{R73Q}$ domain[28] (provided by Takanari Inoue, Johns Hopkins University School of Medicine, Baltimore, USA) were inserted into pGEMHE vector in frame with mCherry and EYFP, the FMN2-C fragment and Sec61-β were subcloned to these two vectors respectively. Capped mRNA was synthesized from linearized plasmid templates by using T7 or T3 mMessage mMachine (Ambion), poly-A tailed with Poly(A) tailing kit and purified with RNeasy MinElute Cleanup Kit (Qiagen). All primers that are used in this study are provided in Supplementary Table 2.

**Microinjection**. GV oocytes were microinjected at room temperature with ~5–10 pl of mRNAs or 20 μM siRNA in MEM-PVP[48] containing IMBX using micromanipulator (IM 300 microinjector, Narishige). After injection, the oocytes were cultured at 37 °C, 5% $CO_2$ in M16 medium containing 0.2 mM IBMX for at least 12 h to allow protein expression. mRNAs used for microinjection were: Mito-AcGFP at final concentration of 600 ng/μl, Por1-mCherry at 700 ng/μl, Sec61β-mKate2 at 650 ng/μl. For Fig. 1b, the mRNA concentration of FMN2-AcGFP was 725 ng/μl, SLD-AcGFP was 750 ng/μl, CLD-AcGFP is 500 ng/μl; for Fig. 1c, FMN2$^{ΔCLD}$ was 825 ng/μl, FMN2$^{ΔSLD}$ was 750 ng/μl. For the dominate negative effect of CLD and SLD, the mRNA concentration of SLD-AcGFP is 2500 ng/μl, CLD-AcGFP was 750 ng/μl. For rescue experiment, the mRNA concentration of

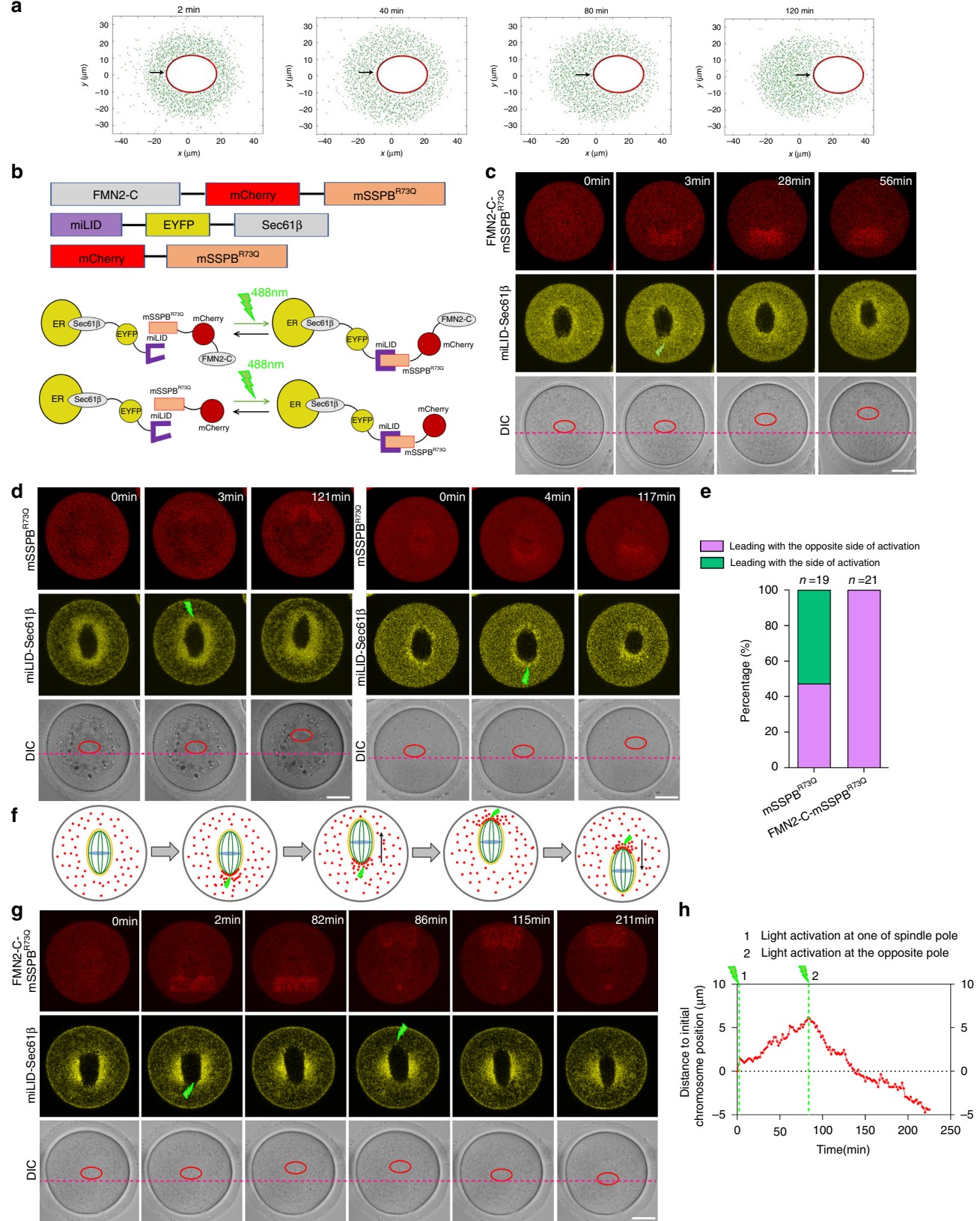

FMN2$^{FL}$-AcGFP was 1450 ng/μl; FMN2$^{ΔCLD}$-AcGFP was 1650 ng/μl; and FMN2$^{ΔSLD}$-AcGFP was 1500 ng/μl, mCherry-myosin Vb tail was 2000 ng/μl, FMN2-C-mCherry-mSSPB$^{R73Q}$ was 830 ng/μl, mCherry-mSSPB$^{R73Q}$ was 850 ng/μl, miLID-EYFP-Sec61β was 900 ng/μl, ATeam1.03-nD/nA was 700 ng/μl, GFP-Rab11a was 800 ng/μl, Mitofusin1-mCherry was 3000 ng/μl, mCherry-myosin Vb tail-Acta was 2000 ng/μl. For the siRNA injection, oocytes were kept in IMBX for 24 h to enable Drp1 knock down. To directly label endoplasmic reticulum, DiIC16

(3) (D384) was microinjected as saturated solution in corn oil (Sigma, C8267) 30 min before imaging.

**Immunoblotting and antibodies**. For immunoblot analysis, a total of 50 oocytes of each group were placed in Laemmli sample buffer (SDS sample buffer and 2-Mercaptoethanol) and boiled at 100 °C for 5 min. The samples were loaded on

**Fig. 5 Asymmetric actin nucleation leads to biased direction of spindle migration. a** A representative of 15 simulations showing each time that if the initial distribution of actin nucleation is asymmetric between the two spindle halves, the spindle will move to the side opposite to the one with more actin nucleation, as a result of more pushing force at the end with more actin nucleation. (The black arrow shows the pole of spindle that has more actin nucleation). **b** Top: constructs used for optogenetic activation of additional actin nucleation at one end of the spindle, the C-terminal half of FMN2 was fused to mCherry-mSSPB$^{R73Q}$ domain, Sec61β fused to miLID-EYFP. Bottom: illumination with 488 nm light leads to miLID domain binding with mSSPB$^{R73Q}$, thus recruiting FMN2-C-mCherry or the control mCherry alone to the ER. **c** Recruitment of FMN2-C-mCherry-mSSPB$^{R73Q}$ to one spindle pole (green strike) by photoactivation of the miLID-EYFP-Sec61β induced the spindle movement with the opposite pole leading. Red ellipses show the positions of chromosomes. Red lines show the initial positions of chromosomes. $n = 21$. Scale bar, 20 μm. **d** Recruitment of the control mCherry-mSSPB$^{R73Q}$ to one spindle pole (green strikes) resulted in roughly equal frequency for each pole to lead the spindle movement. Red ellipses show the positions of chromosomes. Red lines show the initial chromosomes positions. $n = 19$. Scale bar, 20 μm. **e** Quantification of spindle migration direction after recruiting mCherry-mSSPB$^{R73Q}$ or FMN2-C-mCherry-mSSPB$^{R73Q}$ to one spindle pole. $n = 19$ (mCherry-mSSPB$^{R73Q}$), $n = 21$ (FMN2-C-mCherry-mSSPB$^{R73Q}$). **f** Schematics of the experiment in which FMN2-C-mCherry-mSSPB$^{R73Q}$ was recruited to one of the spindle poles by the first light exposure (green strike at 2 min), and after the spindle initiated apparent movement, a second illumination recruited FMN2-C-mCherry-mSSPB$^{R73Q}$ to the opposite pole (green strike at 86 min). **g** A representative montage from the experiment described in **f**, showing the second illumination reversed the direction of spindle migration. Red ellipses show the positions of chromosomes. Red lines show the initial chromosome positions. $n = 5$. Scale bar, 20 μm. **h** A plot of the movement of chromosome position in **g**. Green strike represents the time of two different activations by 488 nm light. Source data are provided as a Source Data file.

4–20% Mini-PROTEAN® TGX™ Gels (Bio-Rad); after electrophoresis, the samples were transferred to polyvinylidene fluoride (PVDF) membranes and blocked for 1 h in TBST containing 5% nonfat milk. Anti-DRP1 (1:1000) was used as primary antibodies (Cell Signaling Technology, #8570), followed by anti-rabbit IgG, HRP-linked antibody (1:5000) (Cell Signaling Technology, #7074). The enhanced chemiluminescence signal was detected using the SuperSignal West Femto kit (Thermo Scientific) and the BioRad ChemiDoc XRS +imager.

**Immunofluorescence or phalloidin staining.** For immunofluorescence staining, oocytes were fixed in 4% paraformaldehyde (30 min) in PBS and then transferred to a membrane permeabilization solution (0.5% Triton X-100 in PBS) for 20 min at room temperature. After blocking with 1% BSA in PBS, the oocytes were stained with anti-DRP1 (1:50) antibody overnight at 4 °C, and then subjected to secondary antibody staining (ThermoFisher Scientific, # SA5-10039, 1:600) for 1 h. For phalloidin staining, oocytes were fixed in 4% paraformaldehyde in PBS at room temperature for 1 h, transferred to a membrane permeabilization solution (0.5% Triton X-100) for 30 min, and then stained with Rhodamine phalloidin (70 nM, Cytoskeleton, cat. #PHDR1) for 2 h. Oocytes were washed three times with 0.1% Tween 20, 0.01% Triton X-100 in PBS, stained with Hoechst 33342 (10 μg/ml in PBS) for 10 min, and then subjected to confocal microscopy imaging.

**Time-lapse imaging and confocal microscopy.** For time-lapse imaging of spindle migration, images were acquired with a confocal microscope (LSM 780; Carl Zeiss) equipped with a C-Apochromat ×40, 1.2 NA water immersion objective. Typically, the whole oocytes were imaged in custom-fabricated microwells by taking a Z-stack of six or seven 2 μm-thick sections every ~2–2.5 min for 3–4 h. Microwells were formed by making a PDMS mold of lithographically generated negative photoresist (SU8-3050; MicroChem Corp., Woburn, MA, USA) features on a silicon wafer. Square microwells measured 500 μm × 500 μm and stood 500 μm tall. For optogenetic activation experiment, the 488-nm laser was used to illuminate the selected region, the images were acquired every 20 s for 3–4 h. laser wavelengths of 514 and 561 nm were used for excitation of EYFP and mCherry, respectively. For immunofluorescence experiment, images were acquired with a Carl Zeiss LSM-800 confocal microscope equipped with C-Apochromat ×63, 1.2 NA oil immersion objective. Images in control and other different treated oocytes were acquired with same imaging conditions.

**Electron microscopy.** For ultrastructural analysis[7,11], jasplakinolide-treated oocytes were fixed with 2% glutaraldehyde in 0.1 M HEPES (pH 7.3), 0.05% saponin, and 0.2% tannicacid (freshly filtered) for 40 min. After washing with 0.1 M HEPES, oocytes were post-fixed with 0.1 OsO4% (aqueous) for 10 min, and En Bloc stained overnight in 0.5% aqueous uranyl acetate. Oocytes were dehydrated through a graded series of ethanol to 100% and embedded in Epon resin. The resin was polymerized at 37 and 60 °C each for 24 h; ultrathin sections (~50–70 nm) were cut on a Leica ultramicrotome (EM UC6; Leica) using diamond knives. Subsequently, the sections were stained with 2% uranyl acetate and lead citrate for 10 and 5 min, respectively. All images were taken on a FEI Technai BioTwin microscope equipped with a Gatan Ultrascan 1000 digital camera. The diameter of different filamentous structures was calculated using pixel resolution and the pixel number across the diameter of the corresponding filaments.

For immunogold labeling, oocytes were fixed for 4–6 h in 4% paraformaldehyde and 0.01% glutaraldehyde in PBS at room temperature, based on our previously published methods[7]. After washing three times in 0.1 M PBS for 15 min each, the samples were dehydrated, infiltrated, and embedded in London resin white resin. 80 nm sections were cut using Leica UC6 ultramicrotome. Then the sections were stained with rabbit-anti-Sec61β for 2 h at room temperature (EMD Millipore, #07-

205, 1:20 dilution), and then incubated with 12-nm gold-conjugated goat anti-rabbit (Jackson laboratories; #111-205-144, 1:20 dilution) for 1 h.

**Mitochondria and ER distribution analysis.** Schematics illustrating the analysis of the mitochondria distribution is shown in Supplementary Fig. 2a. Before quantification of the spindle peripheral mitochondria accumulation, the spindle region of these images was defined by sir-tubulin staining or co-expressed ER maker (Sec61β), and the spindle region was annotated for the next analysis. For each image, the boundaries of the oocyte and spindle region were first manually outlined in image J. The radial distance of each pixels of oocytes in images beyond spindle region area was then computed. These radial distances of pixels were further normalized to values between 0 and 1 to represent the pixel location relative to oocyte boundary (with value of 1) from spindle region (with value of 0) by dividing the radial distances between oocyte boundary and spindle region at each orientation.

The region of spindle peripheral mitochondria was defined by the average intensity curve in wild-type oocytes from the boundary of spindle to oocyte cortex. First, the distance between the peak of average mitochondria intensity to the spindle boundary was found based on the profile of mitochondria intensity, then we took the same distance from the average intensity peak to the other side to find the outer boundary of the spindle peripheral mitochondria layer. The average mitochondria intensity within the spindle peripheral mitochondria region was then calculated. The mitochondria from the outer boundary of spindle peripheral mitochondria layer to oocyte cortex was defined as mitochondria outside spindle periphery and the average intensity of mitochondria in this region was also calculated (see Supplementary Fig. 2a). The same mask was applied to other experimental groups in the same experiments. The fluorescence intensity ratios of mitochondria at spindle periphery to mitochondria outside spindle periphery were compared among different experimental groups.

In Fig. 3m, the mitochondria and ER distribution were quantified by taking the ratio of the intensity of mitochondria and ER at the back of the spindle to the intensity at the front within a 10 μm line in width crossing the long axis of spindle during the spindle migration.

**Tracking of Rab11-labeled endosomes movement.** The images were acquired using a confocal microscope (LSM 780; Carl Zeiss) equipped with a C-Apochromat ×40, 1.2 NA water immersion objective (time interval: 3.87 s). The Rab11a-labeled endosomes that were visible for more than four consecutive timeframes were manually tracked. The speed of Rab11a-labeled endosomes was calculated based on the tracked coordinates.

**Tracking spindle/chromosomes movement.** The spindle/chromosomes movement was tracked by using a Mathematica code. First, the images were applied with Gaussian filter to remove noise. Then the filtered images were binarized according to certain threshold. Some small debris in the binarized images were removed by delete-small-components algorithm. Eventually, the resultant images contained only chromosomes, and the position of chromosomes was defined by component-measurements algorithm of centroid. If there were more than one segment, average position was used. The chromosomes position was traced in each frame according to the location coordinates, and only the displacements projected in the direction pointing from initial to the final location was used, from which the migration speed was computed.

**Modeling spindle migration in oocytes.** We consider the spindle as an elliptical rigid body with long semi-axis $a$ and short semi-axis $b$ (Fig. 4a–c). The position of the center of the spindle is described by $\mathbf{x}_s = (x_s, y_s)$ and the rotation by $\theta_s$, which

are all functions of time. ER is assumed to attach to the surface of the spindle as the spindle moves based on experimental observations. Actin filaments are nucleated from ER and therefore from the surface of the spindle. Denote $\mathbf{x}^i_{sa} = (x^i_{sa}, y^i_{sa})$, $i = 1, ..., N_{fa}$, as the coordinate of contact for each actin filament bundle at the spindle, where $N_{fa}$ is the total number of actin filaments bundles being simulated. The coordinate of the other end of each actin filament bundles is denoted as $\mathbf{x}^i_{am} = (x^i_{am}, y^i_{am})$. While polymerization happens at $\mathbf{x}^i_{sa}$ (barbed end), depolymerization occurs at $\mathbf{x}^i_{am}$ (pointed end). Mitochondria (Mitos) are modeled as rigid spheres with radius $R_{mt}$. The center coordinate of each Mito is denoted as $\mathbf{x}^j_{mt} = (x^j_{mt}, y^j_{mt})$, $j = 1, ..., N_{mt}$, where $N_{mt}$ is the total number of Mitos being simulated.

Actin bundles are balanced under the interaction force, $\mathbf{f}^i_{am}$, from spindle, and the force of polymerization, $\mathbf{f}^i_p$, from spindle. Since actin filaments are relatively thin, here we neglect the viscous drag in the length directions of actin filaments, however, including this drag does not qualitatively change the result (data not shown). Therefore, force balance $\mathbf{f}^i_{am} + \mathbf{f}^i_p = 0$ holds. The magnitude of the interaction force $\mathbf{f}^i_{am}$ is defined as

$$f^i_{am} = |\mathbf{f}^i_{am}| = \begin{cases} k_{am}(d^c_{am} - d^i_{am}) & \text{if } d^i_{am} < d^c_{am} \\ 0 & \text{otherwise,} \end{cases} \quad (1)$$

where $k_{am}$ is the coefficient of the interaction force and $d^i_{am}$ is the distance from the pointed end of the actin filament to the nearest Mito along the direction $\mathbf{n}^i_{fa} = (\mathbf{x}^i_{am} - \mathbf{x}^i_{sa})/|\mathbf{x}^i_{am} - \mathbf{x}^i_{sa}|$ and $d^c_{am}$ is a critical distance for the interaction force to be present. $\mathbf{f}^i_{am} = -f^i_{am}\mathbf{n}^i_{fa}$ by definition. While each actin filament bundle can at most interact with one Mito, each Mito may have multiple forces on it from different actin filament bundles. This $f^i_{am}$ also determines the magnitude of the force of polymerization, i.e., $f^i_p = f^i_{am}$, which in turn is related to the rate of polymerization of each actin filament. Let the length of each filament to be $l_i = |\mathbf{x}^i_{am} - \mathbf{x}^i_{sa}|$, the rate of the change of $l_i$ is governed by a force–velocity relationship for actin polymerization, which has been shown to generally decline with increasing force[49]. Here we take a simplified linear form (nonlinear force–velocity relationship with the same stall force would not change our results):

$$\dot{l}_i = \left[ k_{on}(f^i_p) - k_{off}(f^i_p) \right]\delta, \quad (2)$$

where $\delta$ is the size of a G-actin monomer, and the on- and off-rates of polymerization are

$$k_{on}(f^i_p) = -\frac{K_{on}}{F_{on}}f^i_p + K_{on}, \quad k_{off}(f^i_p) = \frac{K_{off}}{F_{off}}f^i_p, \quad \left( f^i_p < F_{on}, F_{off} \right), \quad (3)$$

where $K_{on}$ and $K_{off}$ are the maximum rates of polymerization and depolymerization, respectively; $F_{on}$ and $F_{off}$ are the cutoff forces of polymerization and depolymerization, respectively. The length of each actin filament is restricted by a maximum persistent length, $l_{fa,max}$.

In addition to forces from actin filaments, each Mito sphere interacts with the rest of Mitos and the spindle through either repulsion or attraction (Fig. 4c). These repulsion and attraction forces may come from the actomyosin network, the fission and fusion processes of mitochondria as discussed, or other unknown factors. Actin polymerization in-between Mitos could also contribute to a fluid-like mitochondrial network. The forces are required for the observed accumulation of mitochondria around the spindle but also allows redistribution of mitochondria relative to the spindle. The magnitude of the potential force is determined by the distance among each pair of objects, i.e.,

$$f_{pot,\alpha} = |\mathbf{f}_{pot,\alpha}| = \begin{cases} -\frac{F^0_{rep,\alpha} + F^0_{atr,\alpha}}{d^0_\alpha}d_\alpha + F^0_{rep,\alpha} & \text{if } d_\alpha \leq d^0_\alpha \\ -\frac{F^0_{atr,\alpha}(d^0_\alpha)^2}{d^2_\alpha} & \text{if } d_\alpha > d^0_\alpha \end{cases}, \quad \alpha \in \{mt, ms\}, \quad (4)$$

where the subscript $\alpha$ represents the relations either between each pair of Mitos ($\alpha = mt$) or between the spindle and each Mito ($\alpha = ms$). $F^0_{rep}$ and $F^0_{atr}$ are the maximal repulsive and attractive forces, respectively. $d^0$ is the transition distance (Fig. 4c). $d$ is the distance between objects under consideration; it is defined as the shortest distance between any two objects (Fig. 4c). Specifically, $d^j_{ms} = |\mathbf{x}^j_{sm} - \mathbf{x}^j_{mt}| - R_{mt}$ is the potential distance between the spindle and the $j$th Mito, where $\mathbf{x}^j_{sm}$ is the closest point on the spindle to the $j$th Mito. $d^{j,k}_{mt} = |\mathbf{x}^k_{mt} - \mathbf{x}^j_{mt}| - 2R_{mt}$ is the distance between each pair of Mitos. The directions of these potential forces are along the distances of $d^j_{ms}$ or $d^{j,k}_{mt}$. The profile of the potential force corresponds to a potential energy, $U$, defined by $f_{pot} = -\nabla U$,

$$U_\alpha = \begin{cases} U^0_\alpha \left[ \left(\frac{d_\alpha}{d^{0'}_\alpha}\right)^2 - \frac{2d_\alpha}{d^{0'}_\alpha} \right] & \text{if } d_\alpha \leq d^0_\alpha \\ -2U^0_\alpha \left[ \left(\frac{d^0_\alpha}{d^{0'}_\alpha}\right)^2 - \frac{d^0_\alpha}{d^{0'}_\alpha} \right] \frac{d^0_\alpha}{d_\alpha} - U^p_\alpha & \text{if } d_\alpha > d^0_\alpha \end{cases}, \quad \alpha \in \{mt, ms\}$$

where

$$\frac{d^0_\alpha}{d^{0'}_\alpha} = \frac{1}{3}\left( 2 + \sqrt{4 - 3\frac{U^p_\alpha}{U^0_\alpha}} \right)$$

The mapping between $(F^0_{rep}, F^0_{atr}, d^0)$ and $(U^0, U^p, d^{0'})$ is

$$F^0_{rep} = 2\frac{U^0}{d^{0'}}, \quad F^0_{atr} = 2\frac{U^0}{d^0}\left[ \left(\frac{d^0}{d^{0'}}\right)^2 - \frac{d^0}{d^{0'}} \right], \quad d^0_\alpha = \frac{1}{3}\left( 2 + \sqrt{4 - 3\frac{U^p_\alpha}{U^0_\alpha}} \right)d^{0'}_\alpha$$

In the model, we can either use $(F^0_{rep}, F^0_{atr}, d^0)$ or $(U^0, U^p, d^{0'})$ to describe the potential force (Supplementary Fig. 5a). In the model for Drp1 knockdown, we reduced the ratio $U^p_{mt}/U^0_{mt}$ by 40%; in the model for myosin Vb tail treatment, we increased the ratio $U^p_{mt}/U^0_{mt}$ by 40%. In both cases, we decreased $d^0_{mt}$' by 23%.

In addition to the potential force, we may also include thermal fluctuation, $f_{R,mt} = |\mathbf{f}_{R,mt}|$, among Mitos. The thermal fluctuation follows a normal distribution with standard deviation $\sigma$,

$$P(f_R) = \frac{1}{\sqrt{2\pi\sigma^2}}\exp\left( -\frac{f^2_R}{2\sigma^2} \right)$$

When $\sigma = 0$, the system is reduced to inter-Mitos forces without thermal fluctuation. Similar to the potential force, the direction of the force from the thermal fluctuation is along the distance of $d^{j,k}_{mt}$. With all the forces considered, for each Mito $j$, the force balance is given by

$$\sum_{i=1}^{n^j_{fa}}\left( -\mathbf{f}^{i,j}_{am} \right) + \sum_{k=1,k\neq j}^{N_{mt}}\mathbf{f}^{j,k}_{pot,mt} + \mathbf{f}^j_{pot,ms} = \eta_{mt}\dot{\mathbf{x}}^j_{mt} \quad (5)$$

where $n^j_{fa}$ is the total number of actin filaments that exert forces on the $j$th Mito. $\eta_{mt}$ is the viscous drag coefficients of Mitos.

To model the spindle, we consider both translational and rotational motions. The forces of polymerization on the actin filaments also act as reaction forces on the spindle. Therefore, the force balances of the translational mode of the spindle are

$$\sum_{i=1}^{N_{fa}}-\mathbf{f}^i_p \cdot \mathbf{n}_a + \sum_{j=1}^{N_{mt}}-\mathbf{f}^j_{pot,ms} \cdot \mathbf{n}_a = \eta_{sa}\dot{\mathbf{x}}_s \cdot \mathbf{n}_a, \quad \sum_{i=1}^{N_{fa}}-\mathbf{f}^i_p \cdot \mathbf{n}_b + \sum_{j=1}^{N_{mt}}-\mathbf{f}^j_{pot,ms} \cdot \mathbf{n}_b = \eta_{sb}\dot{\mathbf{x}}_s \cdot \mathbf{n}_b \quad (6)$$

where we have projected the motion onto the long ($\mathbf{n}_a$) and short ($\mathbf{n}_b$) axes of the spindle and $\eta_{sa}$ and $\eta_{sb}$ are the viscous drag coefficients of the spindle in the two major axes. The rotational motion of the spindle is

$$\sum_{i=1}^{N_{fa}}\mathbf{r}^i_{fa} \times \left( -\mathbf{f}^i_p \right) + \sum_{j=1}^{N_{mt}}\mathbf{r}^j_{mt} \times \left( -\mathbf{f}^j_{pot,ms} \right) = \eta_{sr}\dot{\theta}_s\mathbf{e}_z \quad (7)$$

where $\mathbf{r}^i_{fa} = \mathbf{x}^i_{sa} - \mathbf{x}_s$, $\mathbf{r}^j_{mt} = \mathbf{x}^j_{sm} - \mathbf{x}_s$, and $\eta_{sr}$ is the rotational viscous drag coefficient of the spindle.

The initial condition of the system is set such that the center of the spindle is at the origin with major and minor axes aligned with $x$- and $y$-axis. Actin filaments are randomly distributed around the spindle, each with a random length less than the maximum length, $l_{fa,max}$. In the simulation where the actin filaments are asymmetrically distributed due to optogenetic recruitment, we double the total number of actin filaments, half of which are added to the trailing end of the spindle while the rest half are still randomly distributed. Mitos are distributed outside the actin filaments. The force balances (Eqs. (5)–(7)) are established at each time step. At each time step the length of the actin filaments are checked. When $l_i > l_{fa,max}$ or when the tip of the $i$th actin filament goes into a Mito, the $i$th actin filament is dissolved and a new filament is nucleated at a random location along the spindle with a random length less than $l_{fa,max}$. In the case of asymmetric actin distribution, there is 50% chance that the new actin filament is nucleated at a random location and 50% chance it is nucleated at the trailing end of the spindle. This ensures that roughly half of the actin filaments are randomly distributed and half are located at the trailing end throughout the process.

The model predicts that symmetry breaking of the Mito distribution leads to the spindle migration. To track the co-evolution between the Mito symmetry breaking and spindle migration in the simulations, we plotted the squared displacement of the spindle in the $x$-direction, $|x_s(t) - x_s(0)|^2$, and the Mito distribution together as functions of time (Supplementary Fig. 5b). The Mito distribution was quantified by taking the ratio of the number of Mitos at the back of the spindle to the number at the front within a 10 μm band during the migration (Supplementary Fig. 5c). If the ratio was >1.2, symmetry breaking of the Mito was considered to have started. Likewise, when $|x_s(t) - x_s(0)|^2 > 8$ μm$^2$, the directed spindle migration was deemed to have started. In the sample simulation shown in Supplementary Fig. 5b, symmetry breaking of the Mito started around 97 min and the directed spindle migration starts around 117 min, suggesting a time lag between the two events. We set 210 min as the end point of simulation because this is the typical time scale of the spindle migration process as observed from experiments. Due to the random polymerization of actin and thermal fluctuations in the forces among Mitos, the movements of the spindle and Mitos are not deterministic. For example, for the control case (corresponding to the default set of parameters listed in Supplementary Table 1) the predicted starting time of directed spindle migration varies from simulation to simulation, which is also observed from experiments (Supplementary Fig. 5d). We have quantified the time lag between symmetry breaking of Mitos and the directed migration of spindle both in experiments and in simulations. The model predicts an average delay of 37 min (from 20 simulations),

which matches well with the 35 min experimental observations (from 6 oocytes) (Supplementary Fig. 5e).

Finally, the model predicts that the maximum distance traveled by the spindle is around $a$, the long semi-axis of the spindle. When the spindle migrates away from the original point, (0, 0), it leaves space for the mitochondria to concentrate toward the lagging half. When all the mitochondria have accumulated around the back portion of the spindle, they lose their potential to push the spindle further because the average coordinate of all the mitochondria would remain close to the origin. This is because the average coordinates of the Mitos–spindle system must satisfy

$$\frac{\sum_{i=1}^{N_{mt}} \eta_{mt} x_{mt}(t) + \eta_s x_s(t)}{N_{mt}\eta_{mt} + \eta_s} = 0, \quad \frac{\sum_{i=1}^{N_{mt}} \eta_{mt} y_{mt}(t) + \eta_s y_s(t)}{N_{mt}\eta_{mt} + \eta_s} = 0 \quad (8)$$

for all time $t$. Given the large number of Mitos, we have $N_{mt}\eta_{mt} \gg \eta_s$, suggesting that $\Sigma x_{mt} \approx 0$. Therefore, the maximum distance that the spindle can migrate via this mechanism is its long semi-axis. Note that here the length of the interaction between the spindle and the Mitos, i.e., the length of actin filaments, is negligible compared to the size of the spindle. If the length of interaction is not negligible, then the center of the spindle moves approximately half length of the spindle plus the length of interaction between the spindle and the Mitos.

**Parameter justifications and analysis.** Unless otherwise specified, the default parameters for the model are listed in Supplementary Table 1. These parameters were considered as the control case in simulations where certain parameters were varied. The size of the spindle was estimated from the confocal images of the oocytes. The radius of Mitos was estimated from the EM image in Fig. 3a, from which we also estimated the number density of Mito to be ~1.2/μm². The total area of Mitos around the spindle was estimated from the confocal images. Given the area and density, we obtained the total number of Mitos used in the simulation. The size of the spindle and Mitos also allowed us to estimate their viscous drag coefficients. We estimated the drag coefficients by $\eta = 6\pi\mu r$, where $r$ is the radius of the object and $\mu$ is the dynamic viscosity of the surrounding medium. Since the spindle and Mitos are surrounded by an actin network, the dynamic viscosity of the actin-rich medium was approximated as $\mu = 1$ Pa·s[50,51]. The maximum length of the actin filament, $l_{fa,max} = 0.35$ μm, was estimated from the EM image in Fig. 2a. Single F-actin persistence length has been determined to be around 10 μm. After buckling, the effective length of the actin filament is reduced to <1 μm[52]. In our model, since actin filaments are bundled and the maximum bundle length is only 0.35 μm, they were modeled as rigid rods. The distance between mitochondria and the spindle, $d^0_{ms}$, was obtained from the EM image (Fig. 3a). We measured the distance between each Mito and its adjacent ERs (which is fixed onto the spindle in the model). In total 46 data points were taken and the average distance was calculated to be 0.1 μm.

The maximum forces of actin polymerization and depolymerization were estimated from the force of a single actin filament and the number of actin filaments in the bundle. The average load force of one actin filament is on the order of 1 pN[53]. From the EM image in Fig. 2a there are about 10–20 actin filaments on one layer and the entire bundle may have about 50 filaments. Therefore the maximum forces of actin polymerization and depolymerization were estimated to be 50 pN.

Other than the key parameters associated with the attraction and repulsion profiles (see below), the rest of parameters do not fundamentally change the dynamics of the system. For example, if we double the number of actin filaments used in the model from 160 to 320, the behavior of the model remains the same. We therefore estimate the rest of the non-critical parameters based on their possible relative values to the key parameters and those estimated from experimental observations.

We next analyzed the effect of random forces. Stronger thermal fluctuations, corresponds to larger $\sigma$, tend to break symmetric earlier (Supplementary Fig. 5f). When $\sigma \to 0$, it reduces to a system without thermal fluctuation. In the following, to simplify the matter, we study the impact of each parameter when thermal fluctuation is absent, i.e., $\sigma = 0$. The key parameters in the model that determines the dynamics of the system, i.e., the probable starting time of directed migration of the spindle and the distribution of the Mitos, are the ones that determine the potential profile among Mitos particles. For example, statistically speaking, increasing the ratio $U^p_{mt}/U^0_{mt}$ while fixing $U^0_{mt}$ delays the symmetry breaking (Supplementary Fig. 5g). When the ratio $U^p_{mt}/U^0_{mt}$ is fixed, varying $U^0_{mt}$ does not change the dynamics of the system (Supplementary Fig. 5h). We plotted a phase diagram showing the dynamics of the system with $U^0_{mt}$ fixed and $d^0_{mt}'$ and $U^p_{mt}/U^0_{mt}$ varied. Supplementary Fig. 5i shows the contour of the percentage of cells (out of 15 simulations) that break symmetry within 90 min. The contour shows that larger $d^0_{mt}'$ and smaller $U^p_{mt}/U^0_{mt}$ tend to break symmetry earlier, while smaller $d^0_{mt}'$ and larger $U^p_{mt}/U^0_{mt}$ tend to inhibit symmetry breaking. Based on the correspondence between the potential and the attraction and repulsion forces, this phase diagram indicates that smaller repulsion force or larger ratio of attraction to repulsion tends to break the symmetry at an earlier time.

The effect of attraction and repulsion forces between the spindle and Mitos can be analyzed in a similar way. Since the spindle is propelled by an imbalance of forces from the asymmetric distribution of mitochondria onto the spindle, the repulsion force between the spindle and the mitochondria, $F^0_{rep,ms}$, should be large enough to provide sufficient propulsion force onto the spindle, but also small enough to prevent excessively high speed of the spindle. The attraction force between the spindle and the mitochondria, $F^0_{atr,ms}$, should be several folds smaller than the repulsion force.

In summary, the physics underlying the model ensures that the symmetry breaking happens within a reasonable timeframe for appropriate repulsive and attractive forces among the system.

**Measurements of ATP levels.** Confocal fluorescence lifetime imaging microscopy (FLIM) of oocytes expressing the Förster resonance energy transfer (FRET) sensor ATeam1.03-nD/nA[47] was performed with the use of Zeiss LSM 780 microscope and a PicoQuant system consisting of the PicoHarp 300 time-correlated single photon counting (TCSPC) module, two hybrid PMA-04 detectors, and Sepia II laser control module. The oocytes were maintained in Tecon environmental chamber at 37 °C and 5% $CO_2$. Before FLIM, a single confocal 1024 × 1024 pixel image of the oocytes was acquired using the Apochromat ×40/1.1W Corr 27 lens (Zeiss), excitation with Picoquant 442 nm pulsed diode laser (32.5 MHz, 10% laser intensity at 64% power input) and the Zeiss780 GaAsP detector to capture the CFP and YFP emissions and the transmitted light detector to acquire the DIC images. The FLIM data were collected in the same position in the oocyte while using the same lens, frame size, and laser (100% laser intensity at 64% power input). The emission light was reflected by the 440/485/640 nm triple dichroic mirror (Zeiss) and passed through a transparent plate and the 482 ± 35 nm filter into PicoQuant PMA-04 hybrid detector. The pinhole size was individually set to acquire data from 2 to 3 μm z-sections to limit the emission photon count rate below 10% of the laser excitation rate. The single 1024-pixel square FLIM images were acquired with 58.2 μs pixel dwell time (40 s total scan time).

**FLIM data processing and quantification.** FLIM data were processed with SymPhoTime 64 v2.4 (PicoQuant) software. The data were binned to assure 400–1000 photons per binned pixel, and the three-exponential re-convolution was used to fit the fluorescence decay and the amplitude-weighted average fluorescence lifetime average ($\tau_{AW}$) across entire oocytes. Since changes in FRET efficiency ($E$) are directly linked to changes in donor fluorescence lifetime according to the equation $E = 1 - (\tau_{donor}/\tau_{donorREF})$[54], where the $\tau_{donorREF}$ corresponds to $\tau_{donor}$ measured in the absence of the acceptor, decreased $\tau_{AW}$ of the ATeam1.03-nD/nA sensor signals increased FRET efficiency and higher ATP levels.

**Statistical analysis.** The statistical analysis was performed with GraphPad Prism software 8.0.1 (La Jolla, USA). Comparisions of data was performed by one-way analysis of variance (ANOVA) with Turkey's multiple comparison test or two-tailed unpaired $t$-test, $p$ values < 0.05 was considered significant.

**Reporting summary.** Further information on research design is available in the Nature Research Reporting Summary linked to this article.

## Data availability

Source data underlying Fig. 1f–h, j–l, 2c,d, 3f–h, m, 4d, k–m, 5e, h and Supplementary Figs. 2b, e, g, j, 3b, d, e, 4, 5b, d–i, 6c, d are provided as a Source Data file with the paper. All relevant data supporting the finding of this study are available from the corresponding author upon request. A reporting summary for this article is available as a Supplementary Information file.

## Code availability

The computer codes used for image analysis and the modeling of spindle migration are publicly available on Github at https://github.com/RongLiLab/Duan-et.al.-2019.

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

## Acknowledgements

We thank Marie-Helene Verlhac for providing the mCherry-myosin Vb tail plasmid, Melina Schuh for providing pGEMHE-mCherry-full length myosin Vb plasmid, Philip Leder for providing *Fmn2*-null mice, Jan Ellenberg for providing Fmn2-mCherry, Clare Waterman for providing Fmn2-EGFP. Petr Solc for providing pYX-RNA-eGFP plasmid, Takanari Inoue for providing miLID and mSSPB^R73Q plasmid, and Tom Rapoport for providing the Sec61-β plasmid. We thank Brian Slaughter, Jay Unruh, and Boris Rubinstein for input in data analysis. We thank Jiten Narang and Christopher Lemmon for assistance with microfabrication. This work is supported by grant RO1-HD086577 from the National Institute of Health to RL.

## Author contributions

X.D., K.Y. and R.L. designed experiments; X.D. and K.Y. performed all experiments and analyzed the data except for the electron microscopy which was done by F.G.; Y.L., S.S. constructed the mathematical model, performed numerical simulations, and contributed to image analysis; D.B.M. designed the microfluidics device for oocyte imaging; P.-H.W., P.K., and J.Y. contributed to image analysis; H.Y.W., P.K., E.A.M. and D.W. contributed to experiment design; X.D., Y.L., K.Y., S.S. and R.L. prepared the manuscript; R.L. supervised the study.

## Competing interests

The authors declare no competing interests.
