## [Peer Review File · Nature Communications]

Reviewers' comments:

Reviewer #1 (Remarks to the Author):

The authors propose a model for symmetry breaking in the mouse oocyte, reliant on actin filament pushing by polymerisation on the spindle by using mitochondria as anchors. They combine elegant optogenetic perturbations to biophysical modelling to understand this behaviour quantitatively.

Overall, I find the manuscript and its hypothesis interesting. The general scope of the model is well grounded and justified given the experimental data. The manuscript is well-written and the supplementary material give details on the simulation scheme. However, I do have a number of important concerns about the theory, both on its core assumptions and on the way parameters are related to the data, as right now.

Main points:

1/ Attractive/repulsive forces

The authors state very clearly and honestly the rationale for assuming attractive and repulsive forces in the main text. "As we do not fully understand the processes governing mitochondria morphology and distribution in oocytes, we implemented attractive and repulsive forces between mitochondrial bodies to roughly represent dynamic mitochondrial fusion and fission, respectively. "

However, this choice of modelling still remained slightly cryptic to me. As the ratio of these forces largely dictates the predictions of the model, explaining better what those forces would be is very important, to prevent the model from looking like a "black box" and relate it to biology.

For instance, it would a priori seem more intuitive, when modelling the condition of "reduced mitochondrial fission by knocking down Drp1 ", to simply assume a smaller number of mitochondria (is this in fact the case in addition to the stated mislocalization?). What was the rationale to model it as a change in the attractive force instead? In general, fusion/fission would need to be explained better: for now, it sounds like it's something that should affect primarily mitochondria number, so mapping it to attractive/repulsion forces without good explanations makes the model-data comparison look a bit artificial.

Given the very interesting phenotype reported for Myosin V, I also wondered whether a more minimal model could be implemented: an long-ranged attractive force between spindle and mitosis (i.e. the myosin transport), counterbalanced by diffusion (which in general would be good to estimate theoretically for mitosis of this size and at the time scale of movements). This would reproduce the observations on localisation without the need for instance for attractions between mitosis (would even the repulsion between mitosis be so important in this scenario?).

Because of the number of free parameters, and the unknown around these forces, I think having supplementary figures performing a sensitivity analysis to see the region of phase diagram where the experimentally observed behaviour holds would be important (I realise that the authors do discuss this in two limiting cases in figures, but performing it systematically would make the reading easier).

2/ Parameter estimation

The table 1 provided in Supplementary lists all of the parameters used, but the column "Source" on how they are obtained was empty... (this might have been a conversion issue when making the manuscript). It is therefore hard to exactly understand how and why they are estimated this way.

Some parameters seem easy to estimate based on first principles, for instance the drag coefficient of an ellipsoid or sphere: this only rely on the radius (which the authors measure) and the fluid viscosity (which the authors don't mention, but would be good to have, and discuss explicitly in the supplementary material).

However, there seems to be key parameters which seem fine-tuned without clear justification. For instance, the maximal length of a filament (3.5) is chosen extremely close but superior to the critical distance between repulsion and attraction between mitosis and spindle (3). This seems to be a key ratio, but is not explicitly mentioned, nor is the procedure to estimate both.

Related to point 1: how do the authors estimate the number of mitochondria? was this measured? Seeing whether it's affected in perturbation conditions would also seem important if it's an important parameter of the model.

As a smaller detail: the authors haven't considered the capacity of actin filaments to exert this type of force without buckling (which can occur under their own force). presumably this is because they are bundled, but can the authors use values from the in vitro literature to say a bit more on the

constraints this put? (the parameter values for the maximal force they are using should also be confronted to literature), and to viscous drag forces explicitly in the supplementary.

3/ Symmetry breaking and maximal distance

I have to say that i didn't quite follow the argument in Supplementary as to why spindle migration distance was limited to a critical value, because mitochondria clustered on one side wouldn't give forces "because the average coordinate of all the mitochondria would remain constant"... If there is a pushing force being exerted between spindle and a single mito, this would tend to move one forward and one backward, with the magnitude of each being determined by a respective friction coefficient, wouldn't it? why is the interactions between mitos important in this argument ? I would have thought on the contrary that having a big cluster a interaction mitos on one side would be good, as it's a structure with a lot of friction to push from. But this doesn't seem to be the case in the simulation, so i wonder what i was misunderstanding? (why can't the "average coordinate of all mitochondria" just move in the opposite direction to motion?)

In general, i wondered if a small toy model in 1D (say, of a spindle particle + a cloud of mitos with a spring force to it (or even modelled as a single particle) wouldn't help to understand the mechanisms of symmetry breaking, without as many assumptions and parameters.

4/ Co-evolution of mitochondria asymmetry and directional spindle movement

This is a very interesting observation from the manuscript. Although reproduced in the model, i wonder if the authors would go further in digging the temporal relationship between these events (both in the model and in the experiments). Does the model really predict perfect sync (i could expect some specific delays from one to the other in some parameter region (as in any dynamical system with mutual feedback)? Could this be tested in the data? (for instance measuring dynamic properties of mitochondria distribution as for the spindle in Fig. 5h?)

Reviewer #2 (Remarks to the Author):

Duan et al., investigate the role of FMN2 in initiating migration of the meiotic spindle from the center of the oocyte to the periphery prior to asymmetric cell division. The authors conclude that nucleated actin bundles from ER vesicles are surrounded by mitochondria. The polymerizing actin pushes against the mitochondria and directs the spindle to the periphery thus breaking symmetry.

These well-executed experiments with simple, clear results that elegantly document the role of the SLD domain of FMN2 in spindle migration. My only suggestion for this well-written manuscript would be to carefully copy edit the figure labels for symmetry and separation between words and images.

Reviewer #3 (Remarks to the Author):

The manuscript aims to investigate the mechanisms underpinning actin-mediated symmetry-breaking in mouse oocytes. In pursuing the hypothesis the authors have developed some novel molecular tools to interfere with regional actin polymerisation and as well as optogenetic tools to manipulate actin polymerization. The study concludes that ER-nucleated actin pushes against mitochondria to provide the force necessary to break symmetry. This model favours the hypothesis that pushing forces rather than actin network dynamics is responsible for initiating spindle movement in oocytes. This is a very interesting paper using great techniques but there are a number of questions regarding data interpretation.

Questions and comments:

1. The idea that ER provides the nucleation sites for actin appears to be well characterised in the prior publication and the EM. However, on the Sec6 image in Fig 3, for example, does not show the typical ER network as described by Mehlman, FitzHarris etc using Dil. The Sec6 labelling is not convincing.

Are the authors sure the Sec6 is localising properly, or is there just quite a lot of free Sec6 in the cytosol leading to a high 'background'.

If anything, in these images in Fig 3, the ER appears more focussed at the leading edge of the spindle. A higher concentration of ER would presumably lead to an increased nucleation, which would be sufficient to initiate asymmetry, but in this example at the cortical end, rather than the other. Can ER distribution be quantified?

2. The targeted depletion of Fmn2 mediated actin using SLD has a remarkable effect on mitochondrial localisation. What happens when CCD is used to break down the cytoplasmic network? In the *fmn2*^{-/-} oocytes the mitochondria still seem to get to the spindle – or at least form clusters. I am not clear why local inhibition of actin prevents accumulation while complete inhibition has a very different effect? Are the authors sure about the specificity of the SLD construct – yes it can be reversed but this could be achieved by overexpression - or is there another explanation?
3. A number of other studies have shown that mitochondrial accumulation around the spindle is dependent on MTs and Dynein (Van Blerkom, Dalton more recently). Is SLD disrupting MT-mediated mitochondrial trafficking or having any effect on other cellular activities – motor proteins, for example?
4. Fig 2 lacks quantification across a population of cells.
5. In the discussion the reference to a burst of mtDNA proliferation seems to be in human and I could not find similar findings in mice, at least not without papers saying there is no increase.

Reviewer #4 (Remarks to the Author):

Previous studies revealed a role for actin polymerization by the ER-associated formin FMN2 and the Arp2/3 complex in migration of the meiotic spindle from the center to the periphery of oocytes prior to polar body extrusion. Previous studies from this laboratory revealed that ER and FMN2 that localize in the spindle periphery are critical for this movement. Specifically, they found that inhibition of the localization of ER and its associated FMN2 to the spindle periphery by treatment with nocodazole, inhibited spindle migration. They also found that ER, FMN2 and F-actin which are initially uniformly distributed around the spindle, break symmetry (accumulate on the pole of the spindle that is closer to the cell center). On the basis of these findings they propose that F-actin that is polymerized by FMN2 at the spindle periphery provides part of the pushing force for cortical spindle movement (Yi et al., 2013).

In the manuscript under consideration, the authors provide additional, compelling evidence that spindle associated and not cortical FMN2 is critical for meiotic spindle migration and polymerization of actin around the spindle. They also confirm that ER associate with the spindle periphery and

undergo relocalization to one pole of the spindle. Finally, using optigenetic tools, the authors obtained evidence that promoting actin polymerization at the spindle pole can result in force generation that contributes to cortical spindle movement.

The authors obtained evidence for a role for mitochondria in this process. Previous studies revealed that mitochondria are recruited to the spindle periphery during oocyte maturation and undergo symmetry breaking similar to that observed in ER and actin (Dalton and Carroll, 2013). The authors confirmed this finding. They also find that inhibition of MyoVb or silencing of Drp1 also inhibit spindle migration, and propose that this is due to effects on mitochondria.

The system under investigation is interesting and important. In addition, the studies provide additional evidence that spindle associated ER and FMN2 are critical for spindle migration. Thus, they extend our understanding of this process to some extent. However, critical controls are missing and the links to mitochondria, while interesting, are not studied in sufficient detail to provide mechanistic insight into the role of mitochondria in spindle migration. Finally, the model for mitochondrial function in spindle migration is not supported by the data provided.

1. Critical controls are missing.

a. The authors find that inhibition of MyoVb results in failure to localize mitochondria to the spindle periphery and inhibits spindle migration, but has no effect on ER localization. However, MyoVb has been implicated in driving the movement of other cellular cargos including recycling endosomes. In addition, overexpression of the MyoVb tail appears to result in abnormal aggregation of mitochondria. Therefore, it is not clear that the defect in spindle migration observed when MyoVb is inhibited is due to effects on mitochondrial localization.

b. The authors find that silencing Drp1 results in defects in the morphology of mitochondria at the spindle periphery and failure to localize mitochondria to one spindle pole. However, silencing of Drp1 has diverse effects including inhibition of peroxisome fission and alterations in mitochondrial respiratory activity. It is not clear that the defects in spindle migration observed when Drp1 is silenced is due to effects on relocalization of mitochondria to one pole of the spindle.

2. The authors find that deletion of FMN2 or failure to localize FMN2 to the spindle periphery results in defects in localization of mitochondria to the spindle periphery. The finding that silencing Drp1 does not inhibit localization of mitochondria to the spindle periphery supports the model that FMN2 function in mitochondrial fission is not required for mitochondria localization to the spindle periphery. The manuscript would benefit from experiments that reveal the mechanism FMN2 function in mitochondrial redistribution to the spindle.

3. The authors propose that F-actin which is polymerized by FMN2 at one pole of the spindle push on mitochondria at that site, which in turn generates forces for spindle migration. First, the authors do not provide any evidence for this model. Indeed , it is possible that mitochondrial function in spindle migration is due to their function in ATP production or calcium homeostasis. Second, if mitochondria are serving as a platform for actin-dependent pushing forces on the spindle, it is not clear why localization of mitochondria to one pole of the spindle is essential for force generation.

4. Previous studies revealed that mitochondria associate with the spindle periphery in a manner that is initially symmetric and later asymmetric (Dalton and Carroll, 2013). The paper should be cited.

Response to reviewers

Reviewer #1 (Remarks to the Author):

The authors propose a model for symmetry breaking in the mouse oocyte, reliant on actin filament pushing by polymerisation on the spindle by using mitochondria as anchors. They combine elegant optogenetic perturbations to biophysical modelling to understand this behaviour quantitatively.

Overall, I find the manuscript and its hypothesis interesting. The general scope of the model is well grounded and justified given the experimental data. The manuscript is well-written and the supplementary material give details on the simulation scheme. However, I do have a number of important concerns about the theory, both on its core assumptions and on the way parameters are related to the data, as right now.

Main points:

1/ Attractive/repulsive forces

The authors state very clearly and honestly the rational for assuming attractive and repulsive forces in the main text. "As we do not fully understand the processes governing mitochondria morphology and distribution in oocytes, we implemented attractive and repulsive forces between mitochondrial bodies to roughly represent dynamic mitochondrial fusion and fission, respectively. "

However, this choice of modelling still remained slightly cryptic to me. As the ratio of these forces largely dictates the predictions of the model, explaining better what those forces would be is very important, to prevent the model from looking like a "black box" and relate it to biology.

For instance, it would a priori seem more intuitive, when modelling the condition of "reduced mitochondrial fission by knocking down Drp1 ", to simply assume a smaller numbers of mitochondria (is this in fact the case in addition to the stated mislocalization?). What was the rational to model it as a change in the attractive force instead? In general, fusion/fission would need to be explained better: for now, it sounds like it's something that should affect primarily mitochondria number, so mapping it to attractive/repulsion forces without good explanations makes the model-data comparison look a bit artificial.

We thank the reviewer for raising this important and valid point. Mitochondrial fusion and fission are well-known phenomena characterizing mitochondrial dynamics. Our experimental data show that these dynamic processes affect mitochondrial redistribution and spindle migration. Our model was therefore aimed to capture these characteristics. As the reviewer also acknowledged the challenge of modeling these complex molecular processes explicitly, we viewed mitochondrial particles (Mitos) undergoing fusion and fission effectively as achieving a fluid-like structure with some cohesion while also allowing for separation and diffusion. The simplest model that can capture this set of features is a system of particles with effective short-range attraction and repulsion. Attraction tends to pull particles together, which may mimic mitochondrial fusion, a process that indeed involved the attractive molecular forces exerted through mitofusin oligomerization. Repulsion tends to separate particles, which may model mitochondrial fission, mediated by the constricting dynamin helices. Actin polymerization between Mitos particles may also contribute to these dynamics. Therefore, although our modeling of mitochondrial fission/fusion is

abstract, we feel that it captures the dynamics of mitochondria particles and the physical properties of the structure comprising these organelles. In the revised manuscript, we have extended our explanation and justification of modeling mitochondrial dynamics in this way (Lines 196-202).

In response to reviewer's question on mitochondrial size after Drp1 knockdown, we show that the total mitochondrial mass measured by using mitotracker was not significantly different between the knockdown and control oocytes (see figures below). Based on this information, we performed an additional simulation of the Drp1-knockdown cells by increasing the size (area) of each Mito by a factor of four and reduce the number of Mitos by the same factor (thus conserving total mitochondria), without altering attraction and repulsion forces. This implementation does prevent spindle migration, but also leads to an accumulated distribution of Mitos in the vertical direction, which is not seen in experiments (see figures below). This vertical accumulation may come from the high effective inter-Mitos cohesion as a result of the higher viscous drag, proportional to the size of Mitos. This result suggests that the effects of fusion and fission processes are more than just defining Mitos size, but they may play a direct role in governing mitochondrial distribution.

Given the very interesting phenotype reported for Myosin V, I also wondered whether a more minimal model could be implemented: an long-ranged attractive force between spindle and mitos (i.e. the myosin transport), counterbalanced by diffusion (which in general would be good to estimate theoretically for mitos of this size and at the time scale of movements). This would reproduce the observations on localisation without the need for instance for attractions between mitos (would even the repulsion between mitos be so important in this scenario?).

We thank the Reviewer for this suggestion but reasoned that it would not work. The net force on the spindle depends on the asymmetry in the number of Mitos to the left and right of the spindle (for the sake of simplicity, we can consider a 1-d scenario) within the range of the attractive force. If there is an imbalance in Mitos number or density, then there should be a motion of the spindle. However, from the point of view Mitos, since they only interact with the spindle and otherwise is free to diffuse, their distribution should be symmetric around the spindle. This can be easily checked with a Smulchowski equation description. Therefore, the Mitos distribution around the spindle should be always symmetric if their diffusion is fast when compared with spindle movement (which appears to be the case from our live imaging). Therefore, in this scenario, the spindle would never move.

In our model, the Mitos-Mitos attraction allows the development of an asymmetric accumulation of Mitos on one side of the spindle, which then generates the force. The Mitos-Mitos attraction also sustains the cohesion between Mitos particles after they get pushed, which is consistent with the observation that Mitos accumulate at the opposite end of the moving spindle. On the other hand, if the attractive force between Mitos is too strong, asymmetric Mitos distribution cannot occur, and the spindle does not move, which resembles the case of Drp1 knockdown or mitofusin 1 over-expression.

Because of the number of free parameters, and the unknown around these forces, i think having supplementary figures performing a sensitivity analysis to see the region of phase diagram where the experimentally observed behaviour holds would be important (I realise that the authors do discuss this in two limiting cases in figures, but performing it systematically would make the reading easier).

We have added a thorough parameter sensitivity analysis as well as a phase diagram showing how the dynamics of the system evolves with relative attraction and repulsion forces among Mitos (Supplementary Figure 4, and lines 575-618). In particular, we first show that the dynamic response of the system is not deterministic for a given set of parameters due to the random process associated with actin dynamics: the predicted starting time of directed spindle migration varies from simulation to simulation. This is consistent with experimentally observed variation in the time between spindle assembly and the onset of spindle migration (lines 560-563 and Supplementary Fig. 4d). We then obtained distributions of the starting time of spindle migration for different sets of parameters and plot the phase diagram based on the distribution (lines 603-618; Supplementary Fig. 4e-h). This parameter sensitivity analysis shows that smaller repulsion force or larger ratio of attraction to repulsion tends to break the symmetry at an earlier time.

2/ Parameter estimation

The table 1 provided in Supplementary lists all of the parameters used, but the column "Source" on how they are obtained was empty... (this might have been a conversion issue when making the manuscript). It is therefore hard to exactly understand how and why they are estimated this way.

We have filled out the column to explain the source of each parameter (Supplementary Table 1).

Some parameters seem easy to estimate based on first principles, for instance the drag coefficient of an ellipsoid or sphere: this only rely on the radius (which the authors measure) and the fluid viscosity (which the authors don't mention, but would be good to have, and discuss explicitly in the supplementary material).

We have now added this information in the Methods (Line 582-585). We estimate the drag coefficients by $\eta = 6\pi\mu r$, where r is the radius of the object and μ is the dynamic viscosity of the surrounding medium. Since the spindle and Mitos are surrounded by actomyosin network, the dynamic viscosity of the actomyosin-rich medium is approximated as $\mu = 1 \text{ Pa}\cdot\text{s}$ [1, 2].

However, there seems to be key parameters which seem fine-tuned without clear justification. For instance, the maximal length of a filament (3.5) is chosen extremely close but superior to the critical distance between repulsion and attraction between mitosis and spindle (3). This seems to be a key ratio, but is not explicitly mentioned, nor is the procedure to estimate both.

We apologize for not giving sufficient explanation on the choice of parameters values in previous manuscript. We originally set the maximal length of the actin filament as 3.5 μm because this is a typical persistence length of actin filament (usually on the order of microns but may also go beyond 10 microns in some cases) in mammalian cells. In light of the reviewer comment, we have now re-examined the length of the actin filament from EM images (Fig. 2a) and have accordingly modified the length of actin filament to 0.35 μm (in the modified simulations actin filaments are not visible because of their short lengths). The distance between mitochondria and the spindle, d_{ms}^0 , was also obtained from EM images (Fig. 3a). We measured the distance between each Mito and its adjacent ERs. In total 46 data points were taken and the average distance was calculated to be 0.1 μm . We have added these modifications in Methods (Lines 585-592).

Related to point 1: how do the authors estimate the number of mitochondria? was this measured? Seeing whether it's affected in perturbation conditions would also seem important if it's an important parameter of the model.

The number of Mitos originally used ($N_{mt} = 240$) in the model was the minimum number (in terms of the order of magnitude) needed to ensure a proper timing of symmetric breaking of the system. Any number larger than that is able to generate similar dynamic results. In response to reviewer's question, we have now estimated the number of Mitos from the EM image in Fig. 3a, which shows the number density of Mito is about 1.2 per μm^2 . The total area of Mitos around the spindle can be approximated from the confocal images. With the area and density, we thus obtained the new number of Mitos ($N_{mt} = 1440$) used in the model. We have included the corresponding information in Methods (Lines 577-581).

As a smaller detail: the authors haven't considered the capacity of actin filaments to exert this type of force without buckling (which can occur under their own force). presumably this is because they are bundled, but can the authors use values from the in vitro literature to say a bit more on the constraints this put? (the parameter values for the maximal force they are using should also be confronted to literature), and to viscous drag forces explicitly in the supplementary.

Actin filament buckling has mostly been studied for a single filament with persistence length around 10 μm . After buckling, the effective length of the actin filament is reduced to less than 1 μm [3]. In our system, since actin filaments are bundled and their lengths are short (0.35 μm), we neglect any bending or buckling of the filaments in the model.

The maximum forces of actin polymerization and depolymerization are estimated from the force of a single actin filament and the number of actin filaments in the bundle. The average load force of one actin filament is on the order of 1 pN [4]. From the EM image in Fig. 2a there are about 10-20 actin filaments in

one layer and the entire bundle is estimated to have actin filaments on the order of 50. Therefore the maximum forces of actin polymerization and depolymerization are estimated to be 50 pN.

The choice of viscous drag has also been added (see the response above).

In short, the physics underlying the model ensures that the symmetry breaking happens within a reasonable timeframe for an appropriate range of repulsive and attractive forces within the system.

3/ Symmetry breaking and maximal distance

I have to say that i didn't quite follow the argument in Supplementary as to why spindle migration distance was limited to a critical value, because mitochondria clustered on one side wouldn't give forces "because the average coordinate of all the mitochondria would remain constant"... If there is a pushing force being exerted between spindle and a single mito, this would tend to move one forward and one backward, with the magnitude of each being determined by a respective friction coefficient, wouldn't it? why is the interactions between mitos important in this argument ? I would have thought on the contrary that having a big cluster a interaction mitos on one side would be good, as it's a structure with a lot of friction to push from. But this doesn't seem to be the case in the simulation, so i wonder what i was misunderstanding? (why can't the "average coordinate of all mitochondria" just move in the opposite direction to motion?)

We agree that the Mitos themselves are not a closed system since the spindle is also a part of the system. The average coordinates of the Mitos (mt)-Spindle (s) system satisfy

$$\frac{\sum_{i=1}^{N_{mt}} \eta_{mt} x_{mt}(t) + \eta_s x_s(t)}{N_{mt} \eta_{mt} + \eta_s} = 0, \quad \frac{\sum_{i=1}^{N_{mt}} \eta_{mt} y_{mt}(t) + \eta_s y_s(t)}{N_{mt} \eta_{mt} + \eta_s} = 0$$

for all time t . Given the large number of Mitos, we have $N_{mt} \eta_{mt} \gg \eta_s$, suggesting that $\sum x_{mt} \approx 0$.

Therefore, the maximum distance that the spindle can migrate via this mechanism is its long semi-axis. We have added this explanation (Lines 566-573).

In general, i wondered if a small toy model in 1D (say, of a spindle particle + a cloud of mitos with a spring force to it (or even modelled as a single particle) wouldn't help to understand the mechanisms of symmetry breaking, without as many assumptions and parameters.

Please see the response to Comment #1.

4/ Co-evolution of mitochondria asymmetry and directional spindle movement

This is a very interesting observation from the manuscript. Although reproduced in the model, i wonder if the authors would go further in digging the temporal relationship between these events (both in the model and in the experiments). Does the model really predict perfect sync (i could expect some specific delays from one to the other in some parameter region (as in any dynamical system with mutual feedback)? Could this be tested in the data? (for instance measuring dynamic properties of mitochondria distribution as for the spindle in Fig. 5h?)

The reviewer is correct - asymmetry of the distribution of Mitos (quantified by the number of Mitos at the back and front of the spindle) slightly precedes the directed migration of the spindle (quantified by the squared displacement of the spindle), which is observed both in the numerical simulation and

experimental observation. This is shown in Fig. 3m, Supplementary Figure 3, and Supplementary Fig 4b, described on Lines 181-186 and Lines 548-557.

Reviewer #2 (Remarks to the Author):

Duan et al., investigate the role of FMN2 in initiating migration of the meiotic spindle from the center of the oocyte to the periphery prior to asymmetric cell division. The authors conclude that nucleated actin bundles from ER vesicles are surrounded by mitochondria. The polymerizing actin pushes against the mitochondria and directs the spindle to the periphery thus breaking symmetry.

These well-executed experiments with simple, clear results that elegantly document the role of the SLD domain of FMN2 in spindle migration. My only suggestion for this well-written manuscript would be to carefully copy edit the figure labels for symmetry and separation between words and images.

We appreciate the reviewer's positive comments. We have carefully edited the manuscript during the revision.

Reviewer #3 (Remarks to the Author):

The manuscript aims to investigate the mechanisms underpinning actin-mediated symmetry-breaking in mouse oocytes. In pursuing the hypothesis the authors have developed some novel molecular tools to interfere with regional actin polymerisation and as well as optogenetic tools to manipulate actin polymerization. The study concludes that ER-nucleated actin pushes against mitochondria to provide the force necessary to break symmetry. This model favours the hypothesis that pushing forces rather than actin network dynamics is responsible for initiating spindle movement in oocytes. This is a very interesting paper using great techniques but there are a number of questions regarding data interpretation.

Questions and comments:

1. The idea that ER provides the nucleation sites for actin appears to be well characterised in the prior publication and the EM. However, on the Sec6 image in Fig 3, for example, does not show the typical ER network as described by Mehlman, FitzHarris etc using Dil. The Sec6 labelling is not convincing. Are the authors sure the Sec6 is localising properly, or is there just quite a lot of free Sec6 in the cytosol leading to a high 'background'.

Our previous study [5] showed co-localization of Fmn2 and Sec61 β to ER vesicles by immunoEM. In light of the reviewer's comment, we compared the localization of sec61 β with another well-known ER protein VapA. In the images provided in the new Supplementary Figure 2g, Sec61 β co-localizes with VapA. In the study that the reviewer mentioned, the authors injected the lipophilic dye Dil to the oocytes but did not report the concentration of Dil that they used. In our opinion, this dye may be less specific than the ER proteins that we have used as markers. Dil might also induce ER tubulation, which would impair spindle migration [5], but this was not examined in study using Dil. The "high background" referred to by the reviewer was due to the fact that expression of Sec61 β was low in that particular oocyte. In the revised

manuscript, we replaced it with a new figure (Figure 3I) as well as movie (Supplementary Movie 9), which have higher signal relative to background.

If anything, in these images in Fig 3, the ER appears more focussed at the leading edge of the spindle. A higher concentration of ER would presumably lead to an increased nucleation, which would be sufficient to initiate asymmetry, but in this example at the cortical end, rather than the other. Can ER distribution be quantified?

The spindle in the original Fig. 3I had a slight tilt during migration, and so the back edge of spindle was slightly out of focus, which result in the ER appearing more at the leading edge of the spindle. We have now replaced it with a new movie in which the entire spindle was in focus and it shows the ER distribution to be roughly symmetry during spindle migration (Fig. 3I, Supplementary Movie 9). In addition, we also quantified the ER distribution at the leading edge and back edge of spindle during migration, which showed a slight increased at the back edge of spindle (Fig. 3m). This new analysis is described in the main text, line 177-180.

2. The targeted depletion of Fmn2 mediated actin using SLD has a remarkable effect on mitochondrial localisation. What happens when CCD is used to break down the cytoplasmic network?

We have performed the suggested experiment by breaking down cytoplasmic network with Latrunculin A. We found that inhibiting cytoplasmic actin network by Latrunculin A also disrupted the accumulation of mitochondria around spindle (see in Supplementary Fig. 2d, e).

In the *fmn2*^{-/-} oocytes the mitochondria still seem to get to the spindle – or at least form clusters. I am not clear why local inhibition of actin prevents accumulation while complete inhibition has a very different effect? Are the authors sure about the specificity of the SLD construct – yes it can be reversed but this could be achieved by overexpression - or is there another explanation?

We thank the reviewer for pointing out this potential difference, but mitochondria distribution was not uniform in *Fmn2*^{-/-} or SLD-expressing oocytes, and some clustering of mitochondria in the cytoplasm can be seen in both these oocytes. We have provided two more images in Supplementary Fig 2f, which reflect the variation of mitochondrial clustering from oocyte to oocyte. In addition, SLD-AcGFP and FMN2^{ΔCLD}-AcGFP expressed in wild-type oocyte both localized to spindle periphery (Figure 1b, 1c), supporting the interpretation that the effect of SLD was specific.

3. A number of other studies have shown that mitochondrial accumulation around the spindle is dependent on MTs and Dynein (Van Blerkom, Dalton more recently). Is SLD disrupting MT-mediated mitochondrial trafficking or having any effect on other cellular activities – motor proteins, for example?

We thank the reviewer for pointing out previous studies on mitochondrial distribution. A 1991 paper by Blerkom [6] stated that nocodazole treatment prevented accumulation of mitochondria around the GV, however, only one oocyte with poor image quality (likely imaged on a wide-field microscope rather than confocal, not specified in Materials and Methods) was shown without any quantification. We have repeated this experiment but did not detect disruption of mitochondria accumulation by nocodazole (see figures below). In another study [7], Dalton and Carroll suggested that inhibition of dynein or kinesin-1 before GVBD reduced the accumulation of mitochondria around the spindle, but the reduction was only partial. We have also expressed the dynein inhibitor dynamitin in oocyte before GVBD and found that while it disrupted spindle organization, mitochondria were still concentrated around the spindle (see

figures below). In our study, we show that actin disruption by LatA before GVBD led to dispersed mitochondria. We also further investigated the role of FMN2 in mitochondria distribution by expressing a Fmn2 mutant specifically disrupting actin nucleation in *Fmn2*^{-/-} oocytes. This mutant did not rescue mitochondrial accumulation around the spindle, suggesting that the role of FMN2 in mitochondrial distribution is related to its role in actin nucleation (see Figure 3g and Supplementary Fig. 2c). We also added a result where direct targeting of the dominant negative myosin Vb tail to mitochondrial outer membrane strongly dispersed mitochondria to the cell cortex (Supplementary Fig. 2k). Combined with the observation that expression of myosin Vb tail also disrupted mitochondrial distribution, our data suggest that actin and myosin Vb-mediated transport play a direct role (discussed on Line174-176). However, an involvement of actin does not exclude the possibility that mitochondria are also under the forces of microtubule motors. It is possible that both cytoskeletal systems are involved. We feel that a detailed elucidation of how mitochondrial distribution is established in MI oocytes is beyond the main question addressed in this study (acknowledged on Line 175-176).

4. Fig 2 lacks quantification across a population of cells.

We have performed the quantification of a population of oocytes (see Fig. 2c, d), which supports our conclusion that the effects of the expression of various FMN2 mutants on spindle migration correlated with their ability to support actin polymerization at the spindle periphery.

5. In the discussion the reference to a burst of mtDNA proliferation seems to be in human and I could not find similar findings in mice, at least not without papers saying there is no increase.

We appreciated the reviewer carefully reading and have cited another two papers that showed mtDNA increased during mouse oocyte maturation in this revised manuscript (Line 310).

Reviewer #4 (Remarks to the Author):

Previous studies revealed a role for actin polymerization by the ER-associated formin FMN2 and the Arp2/3 complex in migration of the meiotic spindle from the center to the periphery of oocytes prior to polar body extrusion. Previous studies from this laboratory revealed that ER and FMN2 that localize in the spindle periphery are critical for this movement. Specifically, they found that inhibition of the localization of ER and its associated FMN2 to the spindle periphery by treatment with nocodazole,

inhibited spindle migration. They also found that ER, FMN2 and F-actin which are initially uniformly distributed around the spindle, break symmetry (accumulate on the pole of the spindle that is closer to the cell center). On the basis of these findings they propose that F-actin that is polymerized by FMN2 at the spindle periphery provides part of the pushing force for cortical spindle movement (Yi et al., 2013).

In the manuscript under consideration, the authors provide additional, compelling evidence that spindle associated and not cortical FMN2 is critical for meiotic spindle migration and polymerization of actin around the spindle. They also confirm that ER associate with the spindle periphery and undergo relocalization to one pole of the spindle. Finally, using optogenetic tools, the authors obtained evidence that promoting actin polymerization at the spindle pole can result in force generation that contributes to cortical spindle movement.

The authors obtained evidence for a role for mitochondria in this process. Previous studies revealed that mitochondria are recruited to the spindle periphery during oocyte maturation and undergo symmetry breaking similar to that observed in ER and actin (Dalton and Carroll, 2013). The authors confirmed this finding. They also find that inhibition of MyoVb or silencing of Drp1 also inhibit spindle migration, and propose that this is due to effects on mitochondria.

The system under investigation is interesting and important. In addition, the studies provide additional evidence that spindle associated ER and FMN2 are critical for spindle migration. Thus, they extend our understanding of this process to some extent. However, critical controls are missing and the links to mitochondria, while interesting, are not studied in sufficient detail to provide mechanistic insight into the role of mitochondria in spindle migration. Finally, the model for mitochondrial function in spindle migration is not supported by the data provided.

1. Critical controls are missing.

a. The authors find that inhibition of MyoVb results in failure to localize mitochondria to the spindle periphery and inhibits spindle migration, but has no effect on ER localization. However, MyoVb has been implicated in driving the movement of other cellular cargos including recycling endosomes. In addition, overexpression of the MyoVb tail appears to result in abnormal aggregation of mitochondria. Therefore, it is not clear that the defect in spindle migration observed when MyoVb is inhibited is due to effects on mitochondrial localization.

We thank the reviewer for making this very valid point that inhibition of Myosin Vb could disrupt other cargos including recycling endosomes. We confirmed that Myosin Vb tail indeed associate with GFP-Rab11-labeled endosomes, but surprisingly it did not affect the speed of GFP-Rab11-labeled endosomes in the cytoplasm of MI oocytes (see Supplementary Fig. 2h, j). To more specifically disrupt mitochondria distribution, we directly tethered Myosin Vb tail to mitochondria through a mitochondrial outer membrane binding motif of ActA [8]. This construct did not associate with Rab11 endosomes in the cytoplasm but fully colocalized with mitochondria and strongly dispersed mitochondria to the cortex and blocked spindle migration (see Fig.4j, l, m and Supplementary Fig. 2i, k). We note that there is a pool of Rab11 colocalizing with mitochondria and Myosin Vb tail at the cortex, the significance of which is currently unclear. As this was currently the best we could do to specifically modulate mitochondria

distribution, we acknowledge “...we could not completely rule out the involvement of other organelles whose distribution is also regulated by myosin Vb.” (Line 243-244)

b. The authors find that silencing Drp1 results in defects in the morphology of mitochondria at the spindle periphery and failure to localize mitochondria to one spindle pole. However, silencing of Drp1 has diverse effects including inhibition of peroxisome fission and alterations in mitochondrial respiratory activity. It is not clear that the defects in spindle migration observed when Drp1 is silenced is due to effects on relocalization of mitochondria to one pole of the spindle.

We thank the reviewer for pointing out that silencing of Drp1 may perturb peroxisome distribution based on the previous study by Sukrut C. Kamerkar, et al, 2018 [9]. However, that study did not show a direct role for Drp1 in peroxisome fission using biochemical assays, unlike the extensive data that exist supporting the role of Drp1 in mitochondrial fission. Because peroxisomes co-localize with mitochondria, the observed clustering of peroxisome could be caused by mitochondria clustering. In addition, another study showed that expressing mutant Drp1 affected mitochondria morphology but not other organelles, including peroxisomes [10]. That being said, to address the reviewer’s concern, we have performed a new experiment which showed that overexpression of mitofusin 1 also resulted in tight mitochondria clustering around the spindle and disrupted spindle migration (Fig. 4j, l, m, described on Line 229-236). Because peroxisomes and mitochondria are closely associated, our data certainly does not rule out peroxisomes participating in the mechanical role proposed for mitochondria, but this detail should not affect the principle of force production conveyed by our model.

2. The authors find that deletion of FMN2 or failure to localize FMN2 to the spindle periphery results in defects in localization of mitochondria to the spindle periphery. The finding that silencing Drp1 does not inhibit localization of mitochondria to the spindle periphery supports the model that FMN2 function in mitochondrial fission is not required for mitochondria localization to the spindle periphery. The manuscript would benefit from experiments that reveal the mechanism FMN2 function in mitochondrial redistribution to the spindle.

There is no evidence from us or others that FMN2 plays a role in mitochondrial fission. We apologize if this was a confusion caused by unclear writing in our previous manuscript. In fact, in *Fmn2*^{-/-} oocytes, mitochondria were dispersed rather than being clustered as in Drp1 KD. In the revised manuscript, we expressed a mutant Fmn2 construct (mutation of residues Ile 1215, Arg 1295 and Lys 1371 in the FH2 domain to Ala), which specifically disrupts the actin nucleation activity of FMN2 [11], in *fmn2*^{-/-} oocytes. The mutant FMN2 did not rescue the defect in mitochondria accumulation around the spindle, in contrast to the wild-type FMN2 construct, suggesting that the role of FMN2 in mitochondrial distribution is related to its role in actin nucleation (see Fig. 3g and supplementary Fig. 2c, discussed on line 150-157). Combined with the observation that expression of myosin Vb tail also disrupted mitochondrial distribution, we suggest that actin and myosin V-mediated transport regulate mitochondrial distribution (discussed on Line 163-174), but we feel that a detailed elucidation of how mitochondrial distribution is established in MI oocytes is beyond the main question addressed in this study.

3. The authors propose that F-actin which is polymerized by FMN2 at one pole of the spindle push on mitochondria at that site, which in turn generates forces for spindle migration. First, the authors do not

provide any evidence for this model. Indeed, it is possible that mitochondrial function in spindle migration is due to their function in ATP production or calcium homeostasis.

Our optogenetic experiments shown in Figure 5 were specifically designed to show that F-actin nucleated from FMN2 associated with ER produces the pushing force. In these experiments, we specifically recruited FMN2 actin nucleation domain to ER surface, through light-induced dimerization at one pole of the spindle, and showed that spindle moves with the opposite pole leading. This is fully consistent with FMN2-nucleated actin generating a pushing rather than pulling force (in the latter case, the spindle would have moved in the opposite direction). In addition to this experiment, we also showed: 1) FMN2 is required for the accumulation of actin in the spindle periphery in a region where ER and mitochondria are concentrated, and this role of FMN2 is required for spindle migration; 2) mitochondria move in the opposite direction relative to the spindle and accumulate to the back half of the moving spindle, suggesting mitochondria are experiencing a counter force; and 3) disrupting mitochondrial dynamics, such as using Drp1 KD or overexpression of MFN1 to cause mitochondrial clustering around the spindle, disrupted spindle migration. Given these multiple lines of evidence, we feel that our model is the best so far to describe force production and symmetry breaking to cause spindle translocation. This is better discussed in the revised main text (first two paragraphs of Discussion).

We thank the reviewer for pointing out a possible effect on ATP production and calcium homeostasis due to perturbation of mitochondria fusion and fission dynamics. To test this, we used a FRET-based ATP biosensor to detect the ATP production in Drp1 KD, MFN1 OE, Myosin Vb tail or Myosin Vb tail-ActA expressing oocytes. The result showed that the ATP level was not perturbed in these oocytes (see supplementary Fig. 5c; described on Line 233-235). We also used Fluo-4 calcium indicators to measure calcium concentrations in Drp1 KD, MFN1 OE, Myosin Vb tail or Myosin Vb tail-ActA expressing oocytes. The result showed that the calcium concentration was also not perturbed in these oocytes (see supplementary Fig. 5d; described on Line 239-241). Admittedly, we cannot rule out subtle changes that are not detected by these sensors.

Second, if mitochondria are serving as a platform for actin-dependent pushing forces on the spindle, it is not clear why localization of mitochondria to one pole of the spindle is essential for force generation.

In our model, mitochondria produce the counter forces that push on spindle when actin filaments polymerized from the ER surface reach mitochondria. For the spindle (or any object) to undergo directional movement, the force experienced by the spindle must be asymmetric or vectoral. Therefore, initially when mitochondria are distributed symmetrically around the spindle, the force is also symmetric and the spindle cannot initiate movement, but when mitochondrial distribution becomes asymmetric with an accumulation to the back half of the spindle, the pushing forces on the spindle become vectoral, allowing the spindle to translocate in the direction of the leading pole. On the other hand, movement of the spindle in the leading pole direction further enhances the mitochondrial distribution asymmetry. It is in fact this feedback loop that causes decisive symmetry breaking and underlies the co-emergence of mitochondrial distribution asymmetry and spindle movement (as shown in Fig. 3l, m). This is now explained more clearly in the first two paragraphs of Discussion.

4. Previous studies revealed that mitochondria associate with the spindle periphery in a manner that is initially symmetric and later asymmetric (Dalton and Carroll, 2013). The paper should be cited.

We appreciate this comment and have incorporated the referred paper in the revised manuscript (Line184-185).

References:

- [1] T. Kim, M.L. Gardel, E. Munro, Determinants of fluidlike behavior and effective viscosity in cross-linked actin networks, *Biophys J*, 106 (2014) 526-534.
- [2] K.S. Zaner, T.P. Stossel, Some perspectives on the viscosity of actin filaments, *J Cell Biol*, 93 (1982) 987-991.
- [3] M.P. Murrell, M.L. Gardel, F-actin buckling coordinates contractility and severing in a biomimetic actomyosin cortex, *Proc Natl Acad Sci U S A*, 109 (2012) 20820-20825.
- [4] M.J. Footer, J.W. Kerssemakers, J.A. Theriot, M. Dogterom, Direct measurement of force generation by actin filament polymerization using an optical trap, *Proc Natl Acad Sci U S A*, 104 (2007) 2181-2186.
- [5] K. Yi, B. Rubinstein, J.R. Unruh, F. Guo, B.D. Slaughter, R. Li, Sequential actin-based pushing forces drive meiosis I chromosome migration and symmetry breaking in oocytes, *J Cell Biol*, 200 (2013) 567-576.
- [6] J. Van Blerkom, Microtubule mediation of cytoplasmic and nuclear maturation during the early stages of resumed meiosis in cultured mouse oocytes, *Proc Natl Acad Sci U S A*, 88 (1991) 5031-5035.
- [7] C.M. Dalton, J. Carroll, Biased inheritance of mitochondria during asymmetric cell division in the mouse oocyte, *J Cell Sci*, 126 (2013) 2955-2964.
- [8] G. Guntas, R.A. Hallett, S.P. Zimmerman, T. Williams, H. Yumerefendi, J.E. Bear, B. Kuhlman, Engineering an improved light-induced dimer (iLID) for controlling the localization and activity of signaling proteins, *Proc Natl Acad Sci U S A*, 112 (2015) 112-117.
- [9] S.C. Kamerkar, F. Kraus, A.J. Sharpe, T.J. Pucadyil, M.T. Ryan, Dynamin-related protein 1 has membrane constricting and severing abilities sufficient for mitochondrial and peroxisomal fission, *Nat Commun*, 9 (2018) 5239.
- [10] E. Smirnova, D.L. Shurland, S.N. Ryazantsev, A.M. van der Blik, A human dynamin-related protein controls the distribution of mitochondria, *J Cell Biol*, 143 (1998) 351-358.
- [11] H. Li, F. Guo, B. Rubinstein, R. Li, Actin-driven chromosomal motility leads to symmetry breaking in mammalian meiotic oocytes, *Nat Cell Biol*, 10 (2008) 1301-1308.

Reviewers' comments:

Reviewer #1 (Remarks to the Author):

The authors have addressed a number of my remarks and questions via new analyses and simulations:

- the main and supp texts are now much clearer regarding the underlying assumptions of the model (in particular mito fusion/fission), and the sensitivity analysis helps readability.

- regarding the explanation on why the spindle can only migrate on the length scale of its size: i think i understand better. Maybe the authors could add still that this condition of friction holds because the length scale of interaction (actin filament length) is very small ? (otherwise, if they were long, mitos on a single side could still provide a push on the spindle despite the high friction)? (this is what initially confused me at the sentence "they lose their potential to push the spindle further ...").

- i thank the author on the clarification and additional quantifications on the co-movements of mitos and spindle, which is cool to see both in the data (main figure) and model (supplementary). As a small comment, i think it would still be nice to do a quantification of the delay in the data, especially as it looks qualitatively consistent with the model. The authors have also added some comments on what sets the time of the start of spindle movement, and i think it would enrich the paper to comment similarly at to what sets the delay between mitos and spindle in the model (and to comment on the comparison to the delay in the experimental coevolution of Fig. 3m).

Reviewer #4 (Remarks to the Author):

The authors have addressed many of the concerns raised. However, the mechanism of FMN2 function in spindle migration remains unresolved. Inhibition of localization of FMN2 to the spindle cortex inhibits spindle migration (Fig. 1), actin polymerization around the spindle (Fig. 2) and localization of mitochondria to the spindle periphery (Fig. 3). However, it also appears to have an effect on chromosome alignment. Time lapse imaging of control cells or cells that express dominant negative CLD or FMN2 Δ CLD reveal that chromosomes remain tightly clustered and aligned during spindle migration. However, in cells expressing cortex localized FMN2 (FMN2 Δ SLD), dominant negative SLD and even SLD-AcGFP (which may have dominant negative effects on endogenous FMN2 localization) chromosomes are not in tight clusters or aligned (Fig. 1). Chromosome localization defects are also evident in FMN2 $^{-/-}$ cells, and cells expressing dominant negative SLD and FMN2 Δ SLD in Fig. 2 and Fig. 3 and in cells that overexpress MyoVb tail alone or in the presence of mitochondria targeted MyoVb.

Major concern: Are defects in spindle migration due to force generation at the spindle, or due to defects in chromosome localization (e.g. through general effects on spindle organization or function in chromosome position control)? Indeed, it is possible that FMN2 is critical for spindle function and that this function is required for spindle migration. Alternatively, FMN2 may be required for spindle function in chromosome positional control and in spindle migration. The authors need to determine whether spindle function and chromosome alignment are compromised in cells in which FMN2 is deleted or diverted away from the spindle. If spindle function is compromised under these conditions, the authors need to determine whether FMN2 functions in spindle migration through effects on spindle function as opposed to force generation spindle migration.

2. The EM shows F-actin forming comet-tail like structures on vesicular structures (Fig. 2). The manuscript would benefit from evidence that those vesicular structures are ER? Along those lines, the manuscript would benefit from evidence that the structures labeled as mitochondria are indeed mitochondria. The authors should provide documentation that organelles are ER or mitochondria, or remove EMs from the manuscript.

Reviewer #1 (Remarks to the Author):

The authors have addressed a number of my remarks and questions via new analyses and simulations:

- the main and supp texts are now much clearer regarding the underlying assumptions of the model (in particular mito fusion/fission), and the sensitivity analysis helps readability.

We thank the reviewer for the helpful suggestions.

- regarding the explanation on why the spindle can only migrate on the length scale of its size: i think i understand better. Maybe the authors could add still that this condition of friction holds because the length scale of interaction (actin filament length) is very small ? (otherwise, if they were long, mitos on a single side could still provide a push on the spindle despite the high friction)? (this is what initially confused me at the sentence “they lose their potential to push the spindle further ...”).

The reviewer is correct that the condition holds because the length of the actin filament is neglected. If the length is significantly larger than that observed the spindle can be pushed further. The center of the spindle is approximately half the length of the spindle plus the length of interaction between the spindle and the mitos. We have added this comment in the Methods (Line 592-595).

- i thank the author on the clarification and additional quantifications on the co-movements of mitos and spindle, which is cool to see both in the data (main figure) and model (supplementary). As a small comment, i think it would still be nice to do a quantification of the delay in the data, especially as it looks qualitatively consistent with the model. The authors have also added some comments on what sets the time of the start of spindle movement, and i think it would enrich the paper to comment similarly at to what sets the delay between mitos and spindle in the model (and to comment on the comparison to the delay in the experimental coevolution of Fig. 3m).

We thank the reviewer for the suggestion. We have quantified the delay both in experiments and in the simulation. The experiments showed an average delay of 35 mins and the model showed an average delay of 37 mins. This match is excellent. This new data is shown in (see Supplementary Figure 5e) and we have added an explanation in Methods (Line 579-582).

Reviewer #4 (Remarks to the Author):

The authors have addressed many of the concerns raised. However, the mechanism of FMN2 function in spindle migration remains unresolved. Inhibition of localization of FMN2 to the spindle cortex inhibits spindle migration (Fig. 1), actin polymerization around the spindle (Fig. 2) and localization of mitochondria to the spindle periphery (Fig. 3). However, it also appears to have an effect on chromosome alignment. Time lapse imaging of control cells or cells that express dominant negative CLD or FMN2 Δ CLD reveal that chromosomes remain tightly clustered and aligned during spindle migration. However, in cells expressing cortex localized FMN2 (FMN2 Δ SLD), dominant negative SLD and even SLD-AcGFP (which may have dominant negative effects on endogenous FMN2 localization) chromosomes are not in tight clusters or aligned (Fig. 1). Chromosome localization defects are also evident in FMN2 $-/-$ cells, and cells expressing dominant negative SLD and FMN2 Δ SLD in Fig. 2 and Fig. 3 and in cells that overexpress MyoVb tail alone or in the presence of mitochondria targeted MyoVb.

Major concern: Are defects in spindle migration due to force generation at the spindle, or due to defects in chromosome localization (e.g. through general effects on spindle organization or function in chromosome position control)? Indeed, it is possible that FMN2 is critical for spindle function and that this function is required for spindle migration. Alternatively, FMN2 may be required for spindle function in chromosome positional control and in spindle migration. The authors need to determine whether spindle function and chromosome alignment are compromised in cells in which FMN2 is deleted or diverted away from the spindle. If spindle function is compromised under these conditions, the authors need to determine whether FMN2 functions in spindle migration through effects on spindle function as opposed to force generation spindle migration.

We thank the reviewer for raising this concern, however, with the evidence explained below, defects in chromosome alignment do not prevent spindle migration:

First, several previous studies including our own paper [1-3] showed that migration of MI chromosomes to the cortex is spindle-independent. In these experiments, microtubule inhibitors were added during chromosome migration and chromosome alignment was completely disrupted, yet migration still occurred.

Second, disruption of dynein function by using dynamitin caused gross spindle and chromosome alignment defect, yet spindle migration still happened (see figures below). In addition, in some control oocytes (see below), chromosomes were not properly aligned yet spindle migration occurred normally.

Third, it was shown by Azoury et al [4] and Julien et al [5] that in $Fmn2^{-/-}$ oocytes, the unmigrated spindle undergoes normal chromosome segregation suggesting that there is no gross chromosome alignment issue. However, during spindle migration, chromosomes oscillate and, in some cases even with untreated oocytes, chromosomes do not appear to be perfectly aligned (as shown above). In the revised manuscript, we replaced all the movies and figures to those showing proper chromosome alignment in various mutant background or treatment conditions. In these examples, despite proper

chromosome alignment, spindle migration was still defective (see Fig.1b, c, e, i; Fig.2c, d; Fig 3c, e, j, k; Fig. 4j).

2. The EM shows F-actin forming comet-tail like structures on vesicular structures (Fig. 2). The manuscript would benefit from evidence that those vesicular structures are ER? Along those lines, the manuscript would benefit from evidence that the structures labeled as mitochondria are indeed mitochondria. The authors should provide documentation that organelles are ER or mitochondria, or remove EMs from the manuscript.

We thank the reviewer for raising the concern about the ER and mitochondria structures. The identity of vesicular ER surrounding the spindle and to which FMN2 localizes was validated previously by using immune-gold labeling in EM [1]. In the revised manuscript, we provide additional immune gold labeling EM image to show that endogenous sec61 β localizes to the ER vesicles in the spindle periphery (Supplementary Figure 3d). We also provide a high magnification image showing the typical cristae of mitochondria in structures that we identified to be mitochondria (Supplementary Figure 3c). Unfortunately, because the condition for immune gold labeling of membranous structures is not compatible with the condition for staining actin filaments to be visualized in EM, we cannot do immunogold labeling in the same samples as shown in Figure 2a. This limitation is well known in electron microscopy research community. To provide additional evidence that the Sec61 β stained structures surrounding the spindle are ER, we provide colocalization data of Sec61 β with Vapa (a known ER protein), BFP-KDEL (containing a known ER retention signal), and Dil, a membrane dye used in a previous study to label ER. As shown in Supplementary Figure 3a, b, all three pairs showed high-level colocalization, with Pearson coefficients above 0.8. These data strengthen our conclusion that actin filaments are nucleated from ER vesicles in the spindle peripheral region. Lastly, even if the vesicles were not ER but some unknown organelles, our mechanical model for how spindle moves is still valid.

Reviewer #3 (Remarks to the Author): **Note new reviewer comments are in bold**

The manuscript aims to investigate the mechanisms underpinning actin-mediated symmetry breaking in mouse oocytes. In pursuing the hypothesis the authors have developed some novel molecular tools to interfere with regional actin polymerisation and as well as optogenetic tools to manipulate actin polymerization. The study concludes that ER-nucleated actin pushes against mitochondria to provide the force necessary to break symmetry. This model favours the hypothesis that pushing forces rather than actin network dynamics is responsible for initiating spindle movement in oocytes. This is a very interesting paper using great techniques but there are a number of questions regarding data interpretation.

Questions and comments:

1. The idea that ER provides the nucleation sites for actin appears to be well characterised in the prior publication and the EM. However, on the Sec6 image in Fig 3, for example, does not show the typical ER network as described by Mehlman, FitzHarris etc using Dil. The Sec6 labelling is not convincing. Are the authors sure the Sec6 is localising properly, or is there just quite a lot of free Sec6 in the cytosol leading to a high 'background'.

Our previous study [5] showed co-localization of Fmn2 and Sec61 β to ER vesicles by immunoEM. In light of the reviewer's comment, we compared the localization of sec61 β with another well-known ER protein VapA. In the images provided in the new Supplementary Figure 2g, Sec61 β co-localizes with VapA. In the study that the reviewer mentioned, the authors

injected the lipophilic dye Dil to the oocytes but did not report the concentration of Dil that they used. In our opinion, this dye may be less specific than the ER proteins that we have used as markers. Dil might also induce ER tubulation, which would impair spindle migration [5], but this was not examined in study using Dil. The “high background” referred to by the reviewer was due to the fact that expression of Sec61 β was low in that particular oocyte. In the revised manuscript, we replaced it with a new figure (Figure 3I) as well as movie (Supplementary Movie 9), which have higher signal relative to background.

The images of Sec61 β -labeled ER are fundamentally different from Dil-labeled ER in mouse oocytes. Using Dil, FitzHarris detect peri-nuclear ER and reticular ER in the cytosol. Staining with Dil also reveals recruitment of ER to the perinuclear region prior to spindle migration and accumulation of ER in the “back edge” of the spindle after migration. In contrast, with Sec61 β - or VapA labeling, reticular ER in the cytosol is not evident. In addition, there is some overlap of Sec61 β with VapA in the new supplemental figure. However, they do not co-localize. There are clusters of accumulation VapA in the cytosol that do not co-localize with Sec61 β . The concern that there is cytosolic accumulation of Sec61 β is sound, and remains unaddressed. The authors claim that this is due to the weak signal of Sec61 β is not compelling. Since Sec61 β -mKate is expressed by microinjection of mRNA encoding Sec61 β -mKate into cells that contain endogenous Sec61 β -mKate, Sec61 β -mKate is effectively overexpressed. Therefore, it is possible that excess Sec61 β -mKate is accumulating the cytosol. Indeed, the clusters of VapA that accumulate in the cytosol may also be a consequence of the method used to visualize ER.

As discussed above, in the revised manuscript, we have provided high-resolution images showing that Sec61 β and VapA colocalized around spindle, the quantification data indicated a high degree of colocalization between Sec61 β and VapA. We also compared the localization of Sec61 β with another well-known ER marker containing the ER retention signal KDEL or the reviewer-mentioned lipophilic dye Dil. The images as well as colocalization quantification show that Sec61 β co-localized with KDEL and Dil-labeled ER around the spindle (Supplementary Figure 3a, b). Thus, Dil staining and Sec61 β localization are not fundamentally different. Our data does not rule out the presence of much finer tubular ER structure in the cytosol, which we previously showed to become more prominent when was grossly over-expressed [1]. In addition, also as explained above, our immuno-gold labeling of endogenous Sec61 β further validated the presence of ER vesicles in the spindle periphery.

If anything, in these images in Fig 3, the ER appears more focussed at the leading edge of the spindle. A higher concentration of ER would presumably lead to an increased nucleation, which would be sufficient to initiate asymmetry, but in this example at the cortical end, rather than the other. Can ER distribution be quantified?

The spindle in the original Fig. 3I had a slight tilt during migration, and so the back edge of spindle was slightly out of focus, which result in the ER appearing more at the leading edge of the spindle. We have now replaced it with a new movie in which the entire spindle was in focus and it shows the ER distribution to be roughly symmetry during spindle migration (Fig. 3I, Supplementary Movie 9). In addition, we also quantified the ER distribution at the leading edge and back edge of spindle during migration, which showed a slight increased at the back edge of spindle (Fig. 3m). This new analysis is described in the main text, line 177-180.

The authors provide quantitation of a slight increase in ER in the back edge of the spindle.

However, there is no analysis to determine whether this back edge accumulation is statistically significant.

We have quantified the back edge accumulation of ER at different time points during the spindle migration. The data showed that there is no significant difference of the ER accumulation at the back edge of spindle during the spindle migration (Supplementary Fig. 3e).

2. The targeted depletion of Fmn2 mediated actin using SLD has a remarkable effect on mitochondrial localisation. What happens when CCD is used to break down the cytoplasmic network?

We have performed the suggested experiment by breaking down cytoplasmic network with Latrunculin A. We found that inhibiting cytoplasmic actin network by Latrunculin A also disrupted the accumulation of mitochondria around spindle (see in Supplementary Fig. 2d, e).

The authors provide new data that address concerns raised.

In the *fmn2*^{-/-} oocytes the mitochondria still seem to get to the spindle – or at least form clusters. I am not clear why local inhibition of actin prevents accumulation while complete inhibition has a very different effect? Are the authors sure about the specificity of the SLD construct – yes it can be reversed but this could be achieved by overexpression -or is there another explanation?

We thank the reviewer for pointing out this potential difference, but mitochondria distribution was not uniform in *Fmn2*^{-/-} or SLD-expressing oocytes, and some clustering of mitochondria in the cytoplasm can be seen in both these oocytes. We have provided two more images in Supplementary Fig 2f, which reflect the variation of mitochondrial clustering from oocyte to oocyte. In addition, SLD-AcGFP and *FMN2*_{ΔCLD}-AcGFP expressed in wild-type oocyte both localized to spindle periphery (Figure 1b, 1c), supporting the interpretation that the effect of SLD was specific.

The Reviewer's concern that mitochondrial distribution is different in *Fmn2*^{-/-} compared to SLD-expressing oocytes is sound and largely unaddressed. Mitochondria are small punctate structures that localize throughout the cytosol in SLD-expressing cells. However, in *Fmn2*^{-/-} there is clear clustering or aggregation of the organelle. The authors provide 2 additional images of mitochondria in *Fmn2*^{-/-} cells and state that the differences observed are due to cell to cell variation. However, they did not carry out rigorous analysis of mitochondrial size and cluster formation in WT, with SLD expression, and in *Fmn2*^{-/-} oocytes.

We have quantified the average mitochondrial size in each individual oocyte in WT, SLD-expressing oocytes and *Fmn2*^{-/-} oocytes. the quantification data showed that there is no difference between SLD-expressing oocytes and *Fmn2*^{-/-} oocytes, and the average mitochondria size of each individual oocytes also has largely variation in each group (Supplementary Figure 2g).

3. A number of other studies have shown that mitochondrial accumulation around the spindle

is dependent on MTs and Dynein Van Blerkom, Dalton more recently). Is SLD disrupting MT mediated mitochondrial trafficking or having any effect on other cellular activities – motor proteins, for example?

We thank the reviewer for pointing out previous studies on mitochondrial distribution. A 1991 paper by Blerkom [6] stated that nocodazole treatment prevented accumulation of mitochondria around the GV, however, only one oocyte with poor image quality (likely imaged on a wide-field microscope rather than confocal, not specified in Materials and Methods) was shown without any quantification. We have repeated this experiment but did not detect disruption of mitochondria accumulation by nocodazole (see figures below). In another study [7], Dalton and Carroll suggested that inhibition of dynein or kinesin-1 before GVBD reduced the accumulation of mitochondria around the spindle, but the reduction was only partial. We have also expressed the dynein inhibitor dynamitin in oocyte before GVBD and found that while it disrupted spindle organization, mitochondria were still concentrated around the spindle (see figures below). In our study, we show that actin disruption by LatA before GVBD led to dispersed mitochondria. We also further investigated the role of FMN2 in mitochondria distribution by expressing a Fmn2 mutant specifically disrupting actin nucleation in Fmn2^{-/-} oocytes. This mutant did not rescue mitochondrial accumulation around the spindle, suggesting that the role of FMN2 in mitochondrial distribution is related to its role in actin nucleation (see Figure 3g and Supplementary Fig. 2c). We also added a result where direct targeting of the dominant negative myosin Vb tail to mitochondrial outer membrane strongly dispersed mitochondria to the cell cortex (Supplementary Fig. 2k). Combined with the observation that expression of myosin Vb tail also disrupted mitochondrial distribution, our data suggest that actin and myosin Vb-mediated transport play a direct role (discussed on Line174-176). However, an involvement of actin does not exclude the possibility that mitochondria are also under the forces of microtubule motors. It is possible that both cytoskeletal systems are involved. We feel that a detailed elucidation of how mitochondrial distribution is established in MI oocytes is beyond the main question addressed in this study (acknowledged on Line 175-176).

Since FMN2 has microtubule binding activity, the concern is sound. Indeed, the apparent differences in mitochondrial distribution in Fmn2^{-/-} vs SLD-expressing oocytes may be due to actin-independent Fmn2 function. Mitochondrial morphology appears to be disrupted by nocodazole and dynamitin treatment, despite claims to the contrary by the authors. Analysis of the role of microtubules in mitochondrial distribution in oocytes is beyond the scope of this work. However, analysis of the role for Fmn2 function in MT binding on mitochondrial distribution may strengthen the manuscript and resolve different phenotypes observed in Fmn2^{-/-} and SLD-expressing oocytes.

As we discuss above, our quantification data showed that there is no difference of mitochondrial cluster size between Fmn2^{-/-} and SLD-expressing oocytes. Our previous data was intended to show that after nocodazole and dynamitin treatment, mitochondria remained concentrated around the chromosomes/spindle, and that chromosomes still moves to the cortex, which is consistent with earlier studies demonstrating that spindle migration is microtubule-independent. Whether microtubules have other roles in mitochondrial morphology is beyond the main question addressed in this study.

4. Fig 2 lacks quantification across a population of cells.

We have performed the quantification of a population of oocytes (see Fig. 2c, d), which supports our conclusion that the effects of the expression of various FMN2 mutants on spindle migration correlated with their ability to support actin polymerization at the spindle periphery.

The quantitation shown is of actin distribution in individual cells. The concern that Fig. 2 lacks

quantitation across a population of cells (i.e. in more than 1 cell under each experimental condition) has not been addressed.

The quantitation shown in Fig.2 is indeed across a population of cells, please see the explanation in Figure 2 legend.

5. In the discussion the reference to a burst of mtDNA proliferation seems to be in human and I could not find similar findings in mice, at least not without papers saying there is no increase. We appreciated the reviewer carefully reading and have cited another two papers that showed mtDNA increased during mouse oocyte maturation in this revised manuscript. This concern has been addressed.

References

- [1] K. Yi, B. Rubinstein, J.R. Unruh, F. Guo, B.D. Slaughter, R. Li, Sequential actin-based pushing forces drive meiosis I chromosome migration and symmetry breaking in oocytes, *J Cell Biol*, 200 (2013) 567-576.
- [2] M.H. Verlhac, C. Lefebvre, P. Guillaud, P. Rassinier, B. Maro, Asymmetric division in mouse oocytes: with or without Mos, *Curr Biol*, 10 (2000) 1303-1306.
- [3] B. Leader, H. Lim, M.J. Carabatsos, A. Harrington, J. Ecsedy, D. Pellman, R. Maas, P. Leder, Formin-2, polyploidy, hypofertility and positioning of the meiotic spindle in mouse oocytes, *Nat Cell Biol*, 4 (2002) 921-928.
- [4] J. Azoury, K.W. Lee, V. Georget, P. Rassinier, B. Leader, M.H. Verlhac, Spindle positioning in mouse oocytes relies on a dynamic meshwork of actin filaments, *Curr Biol*, 18 (2008) 1514-1519.
- [5] J. Dumont, K. Million, K. Sunderland, P. Rassinier, H. Lim, B. Leader, M.H. Verlhac, Formin-2 is required for spindle migration and for the late steps of cytokinesis in mouse oocytes, *Dev Biol*, 301 (2007) 254-265.

REVIEWERS' COMMENTS:

Reviewer #4 (Remarks to the Author):

Reviewer #4's comments on the authors' response to Reviewer #2's previous concerns:

1. Major concern: Are defects in spindle migration due to force generation at the spindle, or due to defects in chromosome localization (e.g. through general effects on spindle organization or function in chromosome position control)?

The authors addressed the concern that defects in chromosome alignment may not be the basis for the observed defects in spindle migration. However, defects in chromosome alignment are evident in every condition where spindle migration is inhibited (Fig. 1, 2 and 3) Therefore, the conditions used are also producing defects in chromosome alignment.

The authors should not select for cells where chromosomes happen to be aligned and spindle migration is inhibited for Figures 1-3. Instead, they should revise the text to state that defects in chromosome alignment are observed under all conditions where there are defects in spindle migration and describe the evidence that defects in spindle migration are not due to defects in chromosome alignment.

2. The EM shows F-actin forming comet-tail like structures on vesicular structures (Fig. 2). The manuscript would benefit from evidence that those vesicular structures are ER. Along those lines, the manuscript would benefit from evidence that the structures labeled as mitochondria are indeed mitochondria. The authors should provide documentation that organelles are ER or mitochondria, or remove EMs from the manuscript.

Regarding structures described as mitochondria: Cristae are evident in the EM in Supplemental Fig. 3. Structures that may be cristae are evident in Fig. 2. Is it possible to modulate the contrast in that image to enhance visualization of cristae?

Regarding structures described as ER: ImmunoEM may not be compatible with methods to detect actin by EM. However, if the authors do not have definitive evidence that the vesicular structures that are associated with actin in their EMs are ER, they cannot state that "actin filaments form comet tail-like structure with end-on association with the surface of ER vesicles". Here, there are 2 possible solutions.

a. EMs can be retained in the manuscript. Vesicular structures in the EM in Fig. 2 and 3 should not be labeled ER. The text and figure legend should be revised to state that actin comet tails are associated with vesicular structure that may be ER. The authors should carry out high- or super-resolution imaging of actin and ER to provide direct evidence that comet tails are on ER.

b. EMs and text describing the EM could be removed from the manuscript.

Reviewer #4 (Remarks to the Author):

Reviewer #4's comments on the authors' response to Reviewer #2's previous concerns:

1. Major concern: Are defects in spindle migration due to force generation at the spindle, or due to defects in chromosome localization (e.g. through general effects on spindle organization or function in chromosome position control)?

The authors addressed the concern that defects in chromosome alignment may not be the basis for the observed defects in spindle migration. However, defects in chromosome alignment are evident in every condition where spindle migration is inhibited (Fig. 1, 2 and 3) Therefore, the conditions used are also producing defects in chromosome alignment.

The authors should not select for cells where chromosomes happen to be aligned and spindle migration is inhibited for Figures 1-3. Instead, they should revise the text to state that defects in chromosome alignment are observed under all conditions where there are defects in spindle migration and describe the evidence that defects in spindle migration are not due to defects in chromosome alignment.

We thank the reviewer for the suggestions. We have added this statement in lines 128-133 as "Note that some of the SLD-injected WT oocytes, *Fmn2*^{-/-} oocytes and FMN2^{ΔSLD}-injected *Fmn2*^{-/-} oocytes appeared to have imperfectly aligned chromosomes, however the spindle migration defect was not limited to these oocytes, and even in some wild-type oocytes the chromosomes also misaligned during spindle migration."

2. The EM shows F-actin forming comet-tail like structures on vesicular structures (Fig. 2). The manuscript would benefit from evidence that those vesicular structures are ER. Along those lines, the manuscript would benefit from evidence that the structures labeled as mitochondria are indeed mitochondria. The authors should provide documentation that organelles are ER or mitochondria, or remove EMs from the manuscript.

Regarding structures described as mitochondria: Cristae are evident in the EM in Supplemental Fig. 3. Structures that may be cristae are evident in Fig. 2. Is it possible to modulate the contrast in that image to enhance visualization of cristae?

We thank the reviewer for the suggestion. We indeed modulated the contrast but could not further improve the visualization of cristae.

Regarding structures described as ER: ImmunoEM may not be compatible with methods to detect actin by EM. However, if the authors do not have definitive evidence that the vesicular structures that are associated with actin in their EMs are ER, they cannot state that "actin filaments form comet tail-like structure with end-on association with the surface of ER vesicles". Here, there are 2 possible

solutions.

a. EMs can be retained in the manuscript. Vesicular structures in the EM in Fig. 2 and 3 should not be labeled ER. The text and figure legend should be revised to state that actin comet tails are associated with vesicular structure that may be ER. The authors should carry out high- or super-resolution imaging of actin and ER to provide direct evidence that comet tails are on ER.

We thank the reviewer for this suggestion. We have revised the text and Fig.2a and Fig.3a legends accordingly. The relevant structures are now referred to ER-like vesicles, as explained "...short bundles of actin filaments formed comet tail-like structures with end-on association with the surface of vesicles that are likely to be ER (Fig. 2a), although the fixation condition in this experiment was incompatible with immunogold labeling. We therefore referred to these as ER-like vesicles due this uncertainty." (line 143-145). We have indeed attempted to image ER structures and actin by using the Airyscan module of LSM 880 with Airyscan, but the resolution was insufficient to show the ends of individual actin filaments to be associated with ER surface.

b. EMs and text describing the EM could be removed from the manuscript.